# A HIERARCHICAL LANGUAGE MODEL DESIGN FOR INTERPRETABLE GRAPH REASONING

## ABSTRACT

Large language models (LLMs) have seen an increased adoption for tasks with implicit graphical structures, such as planning in robotics, multi-hop question answering, and knowledge probing. However, despite their remarkable success in text-based tasks, LLMs' capabilities in understanding explicit graph structures remain limited, preventing them from fully replacing Graph Neural Networks (GNNs) in graph-centric applications. In this work, we introduce a Hierarchical Language Model (HLM-G) Design that employs a two-block architecture to effectively capture local and global graph information, significantly enhancing graph structure understanding. Our model achieves a new state-of-the-art in graph understanding, outperforming both GNN and LLM baselines. It demonstrates robustness to variations in graph-descriptive prompts, overcoming a key limitation of existing LLMs. Furthermore, we demonstrate the interpretability of our model using intrinsic attention weights and established explainers. Comprehensive evaluations across diverse real-world datasets, covering node, link, and graph-level tasks, highlight our model's superior generalization capabilities, marking a significant advancement in the application of LLMs to graph-centric tasks.

## 1 INTRODUCTION

Large Language Models (LLMs) (Vaswani et al., 2017; Devlin et al., 2018; Achiam et al., 2023; Chowdhery et al., 2023) have demonstrated impressive generative capabilities, revolutionizing multiple fields, including natural language processing (NLP), computer vision (Wang et al., 2024c; Parashar et al., 2024; Liu et al., 2024b), speech recognition (Fathullah et al., 2024), and cross-modal domains (Wu et al., 2023; Koh et al., 2024). Despite this widespread success, their application to graph tasks remains an emerging area of research (Chen et al., 2024c; Ren et al., 2024; Jin et al., 2023). Unlike linear text data, graph data presents unique challenges due to its non-Euclidean topologies and intricate structures (Jin et al., 2023), making it difficult for LLMs to process these complex relationships effectively. As a result, the adoption of LLMs in graph-centric tasks has been limited, with graph models such as GNNs (Kipf & Welling, 2017; Gilmer et al., 2017) continuing to be the state-of-the-art in this domain.

Applying LLMs to graph tasks presents two key challenges. Firstly, real-world graphs, such as molecules, often consist of complex combinations of features and structures (Qin et al., 2023), such as atoms properties and the bonds between atoms. Although LLMs excel at processing feature-based information due to their strong text comprehension abilities, they often struggle with capturing structural details (Hu et al., 2023). This limitation results in suboptimal performance even on simple graph tasks, such as identifying shortest paths (Guo et al., 2023; Wang et al., 2024a; Fatemi et al., 2023). Consequently, LLMs tend to be effective mainly for node-level tasks, making it challenging to apply them to more complex link and graph-level tasks where understanding long-range structures is crucial (Liu et al., 2023; Wu et al., 2021). Secondly, representing graphs using LLMs presents significant scalability challenges (Zhao et al., 2023; Ye et al., 2023b). Describing a graph node with both feature and structural information, as seen in molecular (Dwivedi et al., 2023), citation (Hu et al., 2020b), or knowledge graphs (Dettmers et al., 2018), often results in lengthy prompts, leading to a sharp increase in computational complexity since the attention mechanism in LLMs scales quadratically with input size. This makes the application of LLMs to large graph-based tasks computationally challenging, necessitating specialized designs. On the other hand, a key advantage of employing LLMs for graph tasks is their ability to process graphs in a human-comprehensible manner,

allowing input in the form of straightforward text descriptions. Since LLMs use a human-readable vocabulary, they offer a natural advantage in interpretability compared to the opaque embeddings utilized by Graph Neural Networks (GNNs) (Binder et al., 2016; Longo et al., 2024; Achtibat et al., 2024). However, no prior work has focused on providing interpretable results that explain the structure of graphs.

To address these challenges, we introduce Hierarchical Language Model for Graphs (HLM-G), a novel framework designed to enhance the graph structure comprehension capabilities of LLMs. Unlike conventional LLMs that apply self-attention across all tokens, HLM-G employs a two-block architecture, comprising a local block and a global block, each with specialized attention masking. This hierarchical structure enables the model to initially capture local information in the lower layers, followed by the integration of global-level information in the upper layers. Our approach not only enhances the model's understanding of graph structures but significantly reduces computational costs, making HLM-G more scalable for large-scale graph tasks. Furthermore, our hierarchical design exhibits increased robustness to variations in graph prompts. We also demonstrate the interpretability of our hierarchical language model with both model intrinsic weights and established explainers. Finally, we conduct comprehensive experiments across seven real-world datasets, encompassing citation networks, knowledge graphs, and molecular graphs. Our results validate HLM-G's ability to generalize effectively across node, link, and graph-level tasks, marking a significant advancement in the application of language models to graph-based tasks.

## 2 BACKGROUND AND RELATED WORK

**Problem Setup.** We denote a graph as $G = (\boldsymbol{A}, \boldsymbol{X}, \boldsymbol{E})$, where $\boldsymbol{A} \in \mathbb{R}^{n \times n}$, $\boldsymbol{X} \in \mathbb{R}^{p \times n}$, and $\boldsymbol{E} \in \mathbb{R}^{q \times m}$ represent the adjacency, node feature, and edge feature matrices, respectively. Here, $n$, $m$, $p$, and $q$ denote the numbers of nodes, edges, node features, and edge features, respectively. Building on these, we describe graph tasks in natural language. For each graph $G_i$, we first construct a sequence $U_i$ that encapsulates the natural language descriptions of $G_i$ covering $\boldsymbol{A}_i$, $\boldsymbol{X}_i$, and $\boldsymbol{E}_i$, coupled with a query $Q_i$ describing the prediction task. Each task is also associated with a true label $y_i \in \mathcal{Y}$. This leads to a dataset of sequences $\boldsymbol{U} = \{(U_1, Q_1, y_1), (U_2, Q_2, y_2), \cdots, (U_N, Q_N, y_N)\}$, where each sequence $U_i = \{u_1, u_2, \cdots, u_{l_i}\}$ and all tokens $u_i$ belong to a vocabulary $\mathcal{V}$.

**LLM Inference Methods.** Prompt engineering has been pivotal in adapting LLMs for a wide range of tasks (Sahoo et al., 2024a; Zhou et al., 2022). Early attempts in prompt engineering for graph tasks involved using structured representations like edge lists and adjacency matrices (Brandes et al., 2013; Zhao et al., 2023), but these struggled with graph structural reasoning tasks (Guo et al., 2023). NLGraph (Wang et al., 2024a) sought to convert graph data into natural language prompts, yet fundamental graph operations remained challenging, even for small graphs. Studies suggest simpler prompts can be more effective, but overall improvements are modest (Zhao et al., 2023; Fatemi et al., 2023; Sahoo et al., 2024b). LLMs continue to underperform compared to specialized graph models, indicating a significant gap (Hu et al., 2023). Beyond prompt engineering other approaches (Yao et al., 2024; Wang et al., 2022) involves exploring multiple reasoning paths and selecting the most confident one, offering marginal gains but at the cost of increased inference time. The limited success of LLMs on graphs has been partly attributed to their inability to construct coherent world models, often relying on pattern matching rather than genuine reasoning (Valmeekam et al., 2023; Stechly et al., 2024).

**LLM Fine-Tuning Approaches.** Fine-tuning and instruction tuning have been investigated to address LLMs' limitations in graph reasoning tasks. Fine-tuning on graph-specific datasets has achieved limited success, with models still struggling to capture complex graph structures (Tang et al., 2023; Vafa et al., 2024). Instruction tuning, which aligns training objectives with graph reasoning tasks, has shown more promise (Wang et al., 2024b; Luo et al., 2024) by introducing a variety of related tasks during training, enabling the LLM to gain a deeper understanding of the graph domain. However, this approach remains labor-intensive and continues to face challenges with large and dense graphs. Methods such as GraphWiz (Chen et al., 2024a) have further incorporated RL preference alignment (Rafailov et al., 2024), demonstrating some improvements but still struggling on dense graph structures. Furthermore, incorporating real-world graph features, such as node and edge attributes found in citation networks, into LLMs remains an open challenge, indicating that more work is needed to fully adapt LLMs for graph tasks.

**Hybrid GNN-LLM Approaches.** Hybrid models aim to leverage the complementary strengths of LLMs and GNNs by combining the textual understanding capabilities of LLMs with the graph-processing proficiency of GNNs. In this approach, LLMs are often used to enhance graph representations by providing enriched feature descriptions, as demonstrated in models like GIANT (Chien et al., 2021) and LM-GNN (Ioannidis et al., 2022). Alternatively, other methods such as G-Retrieval (He et al., 2024), LLaGA (Chen et al., 2024b), and GraphLLM (Chai et al., 2023) employ LLMs as predictors to improve graph reasoning tasks. Despite their effectiveness, these hybrid models inherit certain limitations associated with GNNs, including the issue of oversmoothing (Rusch et al., 2023) and the need for task-specific architecture designs (You et al., 2020), which require different GNN structures for node-, link-, and graph-level tasks. Additionally, these approaches face challenges in interpretability, as they often rely on opaque embeddings, unlike LLM-only methods that provide more intuitive, language token-level interpretations. Such token-level interpretability is inherently more human-understandable and offers clearer insights into the decision-making process

## 3 HIERARCHICAL LANGUAGE MODEL DESIGN

In this section, we introduce our Hierarchical Language Model, designed to effectively capture both the structural and feature-based aspects of graphs. We begin by explaining how graph data can be transformed into natural language descriptions (Section 3.1). Following this, we describe the model's architecture, which is composed of a local block (Section 3.2) for learning local structural information, a pooling layer (Section 3.3) for integrating structural and feature information, and a global block (Section 3.4) for capturing global information. This hierarchical approach not only guides our model to better understand graph structures but also results in computational advantages.

### 3.1 NATURAL LANGUAGE DESCRIPTIONS OF GRAPHS

Following prior works (Guo et al., 2023; Fatemi et al., 2023) that demonstrate the effectiveness of using simpler graph inputs for LLMs, we define a graph-to-text representation $U$ to describe any graph task in natural language. For a graph $G$ characterized by its adjacency matrix $\boldsymbol{A}$, node features $\boldsymbol{X}$, and edge attributes $\boldsymbol{E}$, we construct textual representations capturing both node feature and 1-hop structural information for each node $v_i$ in $G$. These representations are divided into two components: the node feature annotation $U_i^X$ and the node structure annotation $U_i^{AE}$.

**Node Feature Annotation.** Each node can be effectively described in natural language and presented as input to an LLM. The node feature annotation for a node $v_i$, denoted as $U_i^X$, is a natural language sequence that describes the attributes $\boldsymbol{X}_i$ of $v_i$ over a predefined vocabulary $\mathcal{V}$. The template for $U_i^X$ is as follows:

> $U_i^X$ : Node $$ features: $<feature\_1>$: $<content\_1>$; $<feature\_2>$: $<content\_2>$, $\cdots$, $<feature\_p>$: $<content\_p>$.
> **Example 1 (Citation Network):** Node 97 features: Title: A Zero-Knowledge Revocable Credential Verification Protocol Using Attribute-Based Encryption; Abstract: We introduce a zero-knowledge credential verification protocol leveraging Ciphertext Policy Attribute-Based...
> **Example 2 (Molecule):** Node 10 features: Atomic Number: 7; Degree: 2; Formal charge: 5; Number of Hydrogens: 0; Radical electrons: 0; Hybridization: SP2; Aromatic: True; In Ring: False.

**Node Structure Annotation.** The node structure annotation $U_i^{AE}$ captures the structural connections of node $v_i$ within the graph $G$, including its connections to other nodes and the corresponding edge features. This serves as a textual representation of $\boldsymbol{A}$ and $\boldsymbol{E}$. Let $ne(i)_1, ne(i)_2, \ldots, ne(i)_k$ be the indices of $v_i$'s 1-hop neighbors in $G$. The template for $U_{v_i}^{AE}$ is:

> $U_{v_i}^{AE}$ : Node $$ is connected to $<ne(i)_1>$ with $<edge\_feature_1>$, $<ne(i)_2>$ with $<edge\_feature_2>$, $\cdots$, and $<ne(i)_k>$ with $<edge\_feature_k>$.
> **Example 1 (Citation Network):** Node 20 is connected to nodes 10, 14, and 19.
> **Example 2 (Molecule):** Node 11 is connected to nodes 10 and 13 by a double bond,... and to node 27 by a conjugated double bond.

**Task Query.** We define a task-specific query $Q$ to represent the prediction task in natural language. This query is tailored for each prediction scenario, as demonstrated below:

> $Q_G$ : What is the prediction for$\cdots$
> **Example 1:** What is the shortest distance between nodes 0 and 1?
> **Example 2:** Does the molecule inhibit HIV virus replication?

**Graph Task Reformulation.** Any graph-level task can be reformulated using a concatenation of all nodes' respective $U_{v_i}^X$ and $U_{v_i}^{AE}$ along with the task query $Q$. Formally, this representation is given by:

$$f(G) = \text{concat}(U_G, Q_G) = \text{concat}(U_{v_1}^{AE}, U_{v_1}^X, \ldots, U_{v_n}^{AE}, U_{v_n}^X, Q_G),$$

where $v_i \in G$ and concat$(\cdot)$ represents the sequence concatenation operation.

While the feature descriptions are generally standardized, there are multiple ways to describe the structural information of a graph. We explore various prompting strategies in Appendix F.2.

## 3.2 The Local Block

Since language models cannot inherently understand graphs in their natural structure, we introduce a local-to-global guidance approach, where the model first learns strong local features before capturing information at the global graph level. To implement this, we introduce a local block $M_L$ that employs an intra-node attention masking mechanism. This mechanism ensures that, for each node $v_i$, the combined text sequence $(U_{v_i}^{AE}, U_{v_i}^X)$ is processed independently of other nodes, allowing the model to effectively capture node-specific structures and features. Given an input token sequence $H^l \in \mathbb{R}^{n \times d_k}$ at any transformer layer, where $n$ is the total number of tokens across all nodes and $d_k$ is the embedding dimension, we decompose this sequence into segments: $H^l = \{H_1, H_2, \ldots, H_N\}$, with each segment $H_i \in \mathbb{R}^{n_i \times d_k}$ representing the tokens associated with node $v_i$.

The attention mechanism in the local block is then formulated as:

$$\text{Attention}^{(l)}(Q, K, V) = \text{Diag}\left(\text{Attention}^{(l)}(Q_1, K_1, V_1), \ldots, \text{Attention}^{(l)}(Q_N, K_N, V_N)\right),$$

where

$$\text{Attention}^{(l)}(Q_i, K_i, V_i) = \text{Softmax}\left(\frac{(W_Q^{(l)} H_i^{(l-1)})(W_K^{(l)} H_i^{(l-1)})^T}{\sqrt{d_k^{(l)}}}\right)(W_V^{(l)} H^{(l-1)_i}).$$

This block diagonal attention mechanism also provides several computational advantages. Let $n_i^X$ and $n_i^{AE}$ represent the number of tokens corresponding to the feature annotation $U_i^X$ and structure annotation $U_i^{AE}$ for node $v_i$, respectively. The total number of tokens $n$ for the entire graph is given by $n = \sum n_i$, where $n_i = n_i^X + n_i^{AE}$. By employing this block diagonal attention mechanism, we achieve significant computational efficiency compared to traditional full attention approaches. In standard attention, the computational complexity is typically $\mathcal{O}\left((\sum n_i)^2\right)$, which scales quadratically with the total number of nodes, becoming increasingly expensive for larger graphs. In contrast, our block diagonal design reduces the complexity to $\mathcal{O}\left(\sum n_i^2\right)$, resulting in a linear scaling relative to the number of nodes. This improvement substantially enhances efficiency, especially for larger graph-based tasks, making our approach highly scalable.

## 3.3 Pooling Layer

To integrate structural and feature-based information extracted from the graph, we introduce a pooling mechanism. For each node $v_i$, we first derive local embeddings from the hidden states produced by the local block $M_L$. Specifically, the feature-based embedding is obtained as $z_{v_i}^X = \frac{1}{n_i^X} \sum_{j=1}^{l_i} h_j$, where $h_j$ represents the hidden states corresponding to tokens from $U_i^X$. Similarly, the structure-based embedding $z_{v_i}^{AE}$ is obtained from $U_{v_i}^{AE}$ using the same approach. Next, we combine these embeddings through a parameterized pooling operation to produce the final embedding $z_i$ for each node. Formally, given a sample $U$, the pooled embedding is defined as:

$$z_i = \alpha z_{v_i}^{AE} + (1 - \alpha) z_{v_i}^X,$$

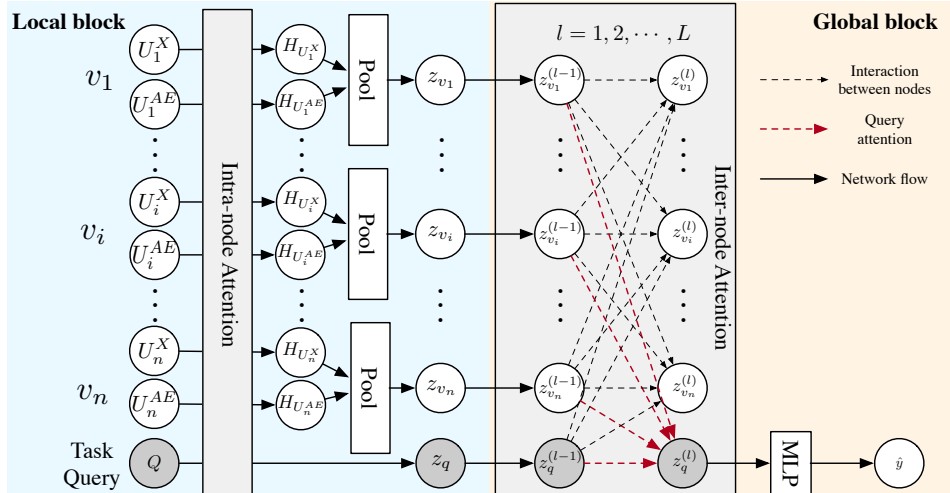

Figure 1: **Hierarchical Model Design**: Local Block employs intra-node attention to learn local node and structural features. Pooling layer combines these features and Global Block utilizes inter-node attention to capture higher-level interactions, enabling comprehensive graph understanding. The Hierarchical model design results in a model which is highly scalable and delivers robust performance across both structure reasoning tasks and real world graph prediction tasks. The model also supports dual interpretability: node-level interpretability through the Global Block and fine-grained token-level interpretability via the Local Block, making it not only powerful but also transparent in its predictions.

where $\alpha \in (0, 1)$ is a trainable parameter that balances the contribution of structural ($\boldsymbol{z}_{v_i}^{AE}$) and feature ($\boldsymbol{z}_{v_i}^{X}$) information. A larger $\alpha$ emphasizes the structural properties in the final prediction, while a smaller $\alpha$ gives more weight to feature-based characteristics.

Our adaptive pooling mechanism allows our model to work for tasks requiring varying levels of structural and feature importance, such as link and graph-level tasks that demand greater structural emphasis and node-level tasks that rely more heavily on feature-based information. We ablate alternative pooling strategies and configurations, which are detailed in Appendix F.

### 3.4 THE GLOBAL BLOCK

To capture global-level interactions across the entire graph, we introduce the global block $M_G$, which leverages a multi-layer transformer architecture to model comprehensive structural relationships. The global block operates on top of the local embeddings derived from $M_L$, learning the higher-level interactions between nodes and enriching the representation with more nuanced graph-level information. Each layer comprises an attention mechanism followed by a feedforward layer. For any layer $l$, the embeddings are updated as:

$$\boldsymbol{Z}^{(l)} = \text{Softmax}\left(\frac{(\boldsymbol{W}_Q^{(l)}\boldsymbol{Z}^{(l-1)})(\boldsymbol{W}_K^{(l)}\boldsymbol{Z}^{(l-1)})^T}{\sqrt{d_k}}\right)(\boldsymbol{W}_V^{(l)}\boldsymbol{Z}^{(l-1)}), \quad \boldsymbol{Z}^{(0)} = [\boldsymbol{z}_{v_1}, \cdots, \boldsymbol{z}_{v_n}, \boldsymbol{z}_q],$$

where $d_k$ is the dimensionality of the key vectors, and $\boldsymbol{W}_Q^{(l)}, \boldsymbol{W}_K^{(l)}, \boldsymbol{W}_V^{(l)}$ are the weight matrices. The input $\boldsymbol{Z}^{(0)}$ includes node embeddings $\boldsymbol{z}_{v_1}, \ldots, \boldsymbol{z}_{v_n}$ and the task-specific query embedding $\boldsymbol{z}_q$ from $M_L$. After processing through $L$ layers, the final embedding $\boldsymbol{z}_q^{(L)}$ is passed through a multilayer perceptron (MLP) to generate the prediction:

$$\hat{y} = \arg\max \text{MLP}(\boldsymbol{z}_q^{(L)}), \quad \text{where } \boldsymbol{z}_q^{(L)} = \boldsymbol{Z}_{:,n+1}^{(L)},$$

with $\hat{y}$ representing the predicted class label. The training objective is to minimize cross-entropy loss:

$$\{\theta_L, \theta_G, \psi\}^* = \operatorname*{argmin}_{\theta_L, \theta_G, \psi} \mathbb{E}_{(U,y)\sim\boldsymbol{U}} \left[\ell(y; \text{MLP}(M_G(M_L(U))))\right],$$

where $y$ is the ground truth label, and $\theta_L, \theta_G, \psi$ represent the trainable parameters of $M_L$, $M_G$, and the MLP, respectively.

Table 1: **Graph reasoning performance comparisons.** This table showcases our HLM-G model against 11 baselines across 7 graph reasoning datasets. Our method not only achieves state-of-the-art performance among all LLMs but also outperforms GNNs on 6 out of 7 tasks. The table details the performance of each method in terms of accuracy across various node, link, and graph-level tasks, underlining the superior capability of our HLM-G model in handling complex graph reasoning challenges with remarkable efficiency and effectiveness.

| Type | Method | Node Degree | Edge Existence | Shortest Distance | Reachable | Cycle | Edge Count | Components |
| | Task level | Node | Link | Link | Link | Graph | Graph | Graph |
| | # Classes | 39 | 2 | 6 | 2 | 2 | 70 | 38 |
| GNN | **GCN** | 7.2±1.61 | 66.5±1.15 | 40.5±2.13 | 87.4±0.99 | 69.1±0.73 | 3.8±0.55 | 8.1±1.94 |
| | **GIN** | 97.7±0.48 | 94.7±0.56 | **96.3±0.16** | 99.9±0.13 | 99.9±0.04 | 65.5±2.35 | 68.8±0.8 |
| | **GTN** | 4.97±0.48 | 50.0±0.48 | 18.5±1.87 | 53.3±0.67 | 50.4±2.3 | 4.7±0.18 | 25.6±0.87 |
| LLM-inference | **Zero Shot** | 15.9±0.00 | 40.8±0.00 | 22.3±0.00 | 34.1±0.00 | 46.4±0.00 | 4.4±0.00 | 1.8±0.00 |
| | **COT** | 37.4±0.00 | 67.2±0.00 | 22.8±0.00 | 34.6±0.00 | 23.8±0.00 | 4.1±0.00 | −±0.00 |
| | **COT-SC** | 37.9±0.00 | 69.9±0.00 | 24.3±0.00 | 41.8±0.00 | 24.8±0.00 | 7.4±0.00 | −±0.00 |
| | **NLGraph** | 20.4±0.00 | 49.3±0.00 | 13.2±0.00 | 32.4±0.00 | 47.7±0.00 | 0.37±0.00 | 0.55±0.00 |
| Hybrid GNN-LLM | **GraphToken** | 22.4±2.30 | 64.7±0.90 | 54.7±1.34 | 54.6±2.89 | 73.4±1.85 | 7.8±0.31 | 5.2±0.09 |
| LLM-finetuning | **BERT** | 21.7±1.39 | 55.9±2.41 | 61.6±1.34 | 76.0±0.56 | 91.4±0.31 | 97.2±0.26 | 29.2±0.59 |
| | **Llama 3** | 41.1±0.13 | 92.6±1.01 | 48.3±0.34 | 84.7±0.69 | 89.8±0.98 | 29.1±3.16 | 9.2±1.44 |
| | **Graphwiz** | 29.6±1.31 | 87.7±1.11 | 47.1±1.77 | 75.9±1.06 | 84.1±0.09 | 37.8±3.10 | 19.9±3.88 |
| | **HLM-G (Ours)** | **99.9±0.04** | **100±0.00** | 84.6±0.43 | **99.9±0.07** | **99.9±0.06** | **98.6±0.03** | **94.2±0.21** |

# 4 EXPERIMENTS

In this section, we conduct experiments to investigate four specific research questions (RQs) to assess the effectiveness of our model on graph tasks: **RQ1**: Can our model accurately understand the underlying structures and maintain robust performance across different graph reasoning datasets? **RQ2**: Does our approach enhance interpretability performance and produce intrinsic interpretable results? **RQ3**: Can the proposed method handle complex real-world datasets with diverse node or edge features? **RQ4**: Does the proposed method work well across all node, link and graph level tasks?

## 4.1 STRUCTURE UNDERSTANDING CAPABILITIES OVER GRAPH REASONING DATASETS

To answer RQ1, we aim to validate whether our model can process graph structure information by conducting the following experiments on graph reasoning datasets.

**Datasets.** First, following Wang et al. (2024a), we create a synthetic dataset consisting of seven graph reasoning tasks to assess the structural reasoning capabilities of our model. These datasets were constructed by a Random Graph Generator capable of generating graphs with up to 40 nodes and 700+ edges. Further information on these datasets is provided in Appendix C.1.1.

**Baselines.** We compare our method against both GNN-based and LLM-based approaches. On the GNN side, our comparisons include models such as GCN (Kipf & Welling, 2017), GAT (Veličković et al., 2017), and the more expressive GIN (Xu et al., 2018), as well as the graph transformer model, GTN (Yun et al., 2019). For LLMs, our inference-only methods include Zero-Shot (Huang et al., 2023), Chain of Thought (CoT) (Wei et al., 2023), CoT Self Consistency (CoT-SC) (Wang et al., 2022), and Natural Language Graph (NLGraph) (Wang et al., 2024a) prompting. Additionally, fine-tuning baselines such as BERT (Devlin et al., 2019) and Lora-Trained (Hu et al., 2021) Llama 3 are used for direct comparisons. We include GraphWiz (Chen et al., 2024a) as a representative of instruction tuning. The GraphToken (Perozzi et al., 2024) method, which utilizes a GNN encoder to fine-tune a frozen LLM, is also compared. Detailed information on our experimental configurations and hyperparameters can be found in Appendix C.2. For LLMs, we use Llama-3 8B (Dubey et al., 2024) as the primary backbone.

### 4.1.1 QUANTITATIVE COMPARISONS.

Our method demonstrates state-of-the-art performance across all graph reasoning datasets, significantly outperforming all baselines, both GNN and LLM-based models. Notably, GNNs such as GIN, despite their theoretically strong expressiveness as validated by the WL-1 test (Huang & Villar, 2021), struggle with graph-level tasks, failing to match the comprehensive understanding offered by our model. Prompt engineering approaches, like CoT and NLGraph and exploration based ap-

Table 2: **Structural Robustness Assessment.** This table displays the accuracy drop observed over 10 permutations for each task. Lower performance drop (↓) indicates less sensitivity to node description positions, highlighting the model's ability to learn graph structure effectively.

| Method | Node Degree(↓) | Edge Existence(↓) | Shortest Distance(↓) | Reachable(↓) | Cycle(↓) | Edge Count(↓) | Components(↓) |
|---|---|---|---|---|---|---|---|
| NLGraph | 46.1 | 38.1 | 56.6 | 44.1 | 49.4 | 71.6 | 71.3 |
| BERT | 71.4 | 11.0 | 46.2 | 14.3 | 9.4 | 7.8 | 62.8 |
| LLaMA 3 | 21.5 | 11.9 | 28.9 | 8.6 | 15.9 | 44.8 | 62.2 |
| GraphWiz | 18.6 | 23.5 | 32.1 | 15.2 | 26.9 | 38.3 | 42.0 |
| HLM-G (our method) | **0.0** | **0.0** | **6.1** | **0.8** | **0.1** | **3.0** | **10.2** |

proaches like CoT-SC do not yield substantial improvements in performance, particularly on tasks like shortest distance that demand a deeper comprehension of the graph structure. GraphToken, a hybrid GNN-LLM approach shows limited gains, indicating that using GNN-augmented LLMs is insufficient for achieving top performance in graph tasks.

While instruction-tuned models like GraphWiz exhibit better results on smaller graphs, they face significant challenges with larger and denser graphs. Notably, their performance is strong on graphs with up to 100 edges, reaching accuracies of 93% and 84% for the reachable and edge count tasks, respectively. However, this accuracy drops sharply to 76% and 38% when the graphs become denser, with up to 700+ edges, as shown in Table 1. Our model remains highly effective in these dense scenarios, maintaining near-perfect accuracies across all tasks, demonstrating its robustness against graph complexity and density. Fine tuning a similar sized BERT and even 80X larger models like Llama-3 is unable to outperform our architecture, underscoring the fact that our design is better suited for graph based tasks than the traditional design.

### 4.1.2 EVALUATION OF MODEL ROBUSTNESS.

A critical question emerges from the quantitative comparisons: *Do language models truly understand graph structures, or do they rely on pattern-matching?* To investigate this, we conducted a robustness evaluation by systematically shuffling the node indices of each graph using a permutation matrix $P$. Unlike GNNs, which are inherently invariant to changes in node indexing due to their symmetrical message-passing framework, LLMs may exhibit sensitivity to even slight alterations in node token representations, potentially leading to inconsistent predictions for the same graph described differently. This issue highlights a significant shortcoming of LLMs in graph-based tasks.

In this experiment, we applied the permutation matrix $P$ 10 times to each graph, generating modified adjacency matrices $A^t = PA^{t-1}P^T$ at each iteration $t$. This process preserves the overall graph structure while changing the node indices, allowing us to evaluate whether the model's predictions remain consistent under different representations.

The results, presented in Table 2, highlight a stark difference between traditional LLM-based models and our proposed HLM-G model. NLGraph's performance dropped significantly, indicating that prompt engineering is not robust. Similarly, we observed that fine-tuned LLMs, such as Llama 3 and BERT, exhibited performance drops of up to 21% and 71%, respectively, on the Node Degree task. This highlights their high sensitivity to changes in node tokens and suggests a reliance on pattern recognition rather than a true comprehension of the underlying graph structure. Instruction tuning does not seem to provide robustness as Graphwiz also shows similar sensitivity as finetuned Llama 3. In contrast, our HLM-G model displays exceptional robustness, with minimal performance drops (e.g., a mere 6.1% drop on the Shortest Distance task and 0.0% on the Node Degree task). These findings underscore a crucial advantage of our HLM-G, while conventional LLMs struggle with variations in graph representation, our model remains robust, reinforcing its suitability for real-world graph tasks where representations might vary but the underlying structure remains unchanged.

### 4.2 INTERPRETABILITY COMPARISONS

Having established the performance and robustness of our model, we now delve into analyzing its interpretability—specifically, its ability to accurately identify and prioritize the most critical structural elements within graph reasoning tasks, thus addressing RQ2. Interpretability serves as a vital criterion in evaluating whether a model is capable of comprehending graph structures rather than just fitting patterns. For this purpose, we utilize four graph reasoning datasets that offer explicit ground truths

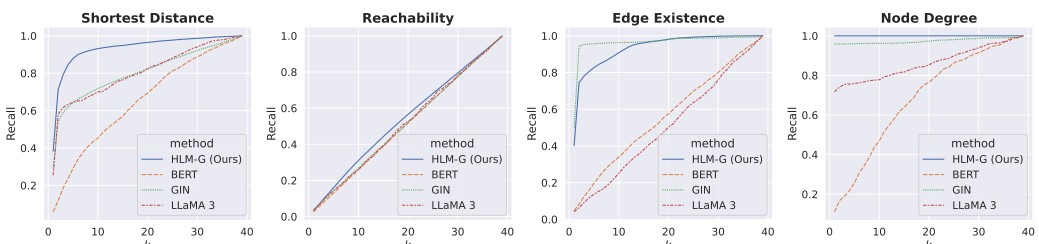

Figure 2: **Explainer Based Interpretation Comparisons.** This figure illustrates the interpretability performance of BERT, GIN, and our method on 4 graph reasoning datasets with reasoning ground truths. $k$ indicates the $k$ most important nodes that interpreted by the model are selected.

regarding which nodes are genuinely important for a given graph task. For example, in the shortest distance task, the ground truth consists of nodes that lie along the shortest path between two specified nodes. More details about these ground truths are provided in Appendix E.1. We compare the true structure understanding capabilities of four finetuned models from Table 1: Llama-3, BERT, GIN and HLM-G.

To measure interpretability, each model is expected to generate an ordered set $\boldsymbol{r} = \{r_1, \ldots, r_n\}$ for a graph with $n$ nodes, ranking them from the most to the least significant, based on the model's internal focus and reasoning. The quality of a model's interpretation is then evaluated by how effectively it identifies the nodes that align with the ground truths. Ideally, a model with true structural comprehension should consistently rank ground truth nodes higher, indicating that it genuinely understands the critical elements of the graph structure. Using established explainers, we first reveal the extent to which our approach successfully captures and prioritizes the essential graph components. We then introduce the intrinsic interpretability mechanism built into our model, demonstrating its ability to provide ready made interpretations.

### 4.2.1 EXPLAINER-BASED INTERPRETATION

To objectively compare the interpretability performance across different models, we leverage established explainability techniques such as Saliency (Simonyan et al., 2013), Input x Gradient (Shrikumar et al., 2016), DeepLIFT (Shrikumar et al., 2017), and GNNExplainer (Ying et al., 2019). This approach allows us to assess how well each model can identify and rank important graph elements, providing insight into the structural modeling capabilities of these models. More details on this strategy, referred to as "explanations as interpretations", are outlined in Appendix E.2.

**Setup.** To quantify interpretability, we use a Recall@$k$ metric, which measures how effectively a model identifies the nodes that correspond to ground truths. Given a set of ground truth nodes $\boldsymbol{r}^{gt}$ and the set of top-$k$ nodes identified by the model $\boldsymbol{r}^k = \{r_1, \ldots, r_k\}$, we calculate Recall@$k$ as $\text{Recall}(k) = \frac{|\boldsymbol{r}^k \cap \boldsymbol{r}^{gt}|}{|\boldsymbol{r}^{gt}|}$, where $|\cdot|$ represents the cardinality of the set and $\cap$ denotes the intersection. As shown in Figure 2, we evaluate each model's performance by plotting the recall curve for $k \in \{1, 2, \ldots, n\}$, where $n$ represents the total number of nodes in the graph. Ideally, a model with a strong understanding of graph structure will have high recall values across different values of $k$, indicating that it consistently identifies the most important nodes.

**Results and Analysis.** Figure 2 presents the interpretability results for the four models across the four graph reasoning datasets. Our HLM-G model demonstrates superior interpretability, particularly as $k$ increases, indicating a higher proficiency in pinpointing the most relevant nodes for each task. While GIN performs adequately on tasks requiring simpler one-hop reasoning, such as Edge Existence and Node Degree, it struggles with more complex, multi-hop reasoning tasks like Shortest Distance and Reachability. In contrast, BERT and LLaMA consistently fail to identify relevant structural features, reflecting their limited capability to capture intricate graph patterns. Directly fine-tuning LLMs has not led to significant improvements in these cases. Although Llama 3 outperforms BERT on three out of four tasks, it still does not reach the performance level of GIN or our model. Our model, in fact, excels across all tasks, even those involving multi-hop reasoning, which further confirms its strong understanding of graph structures beyond simple pattern matching.

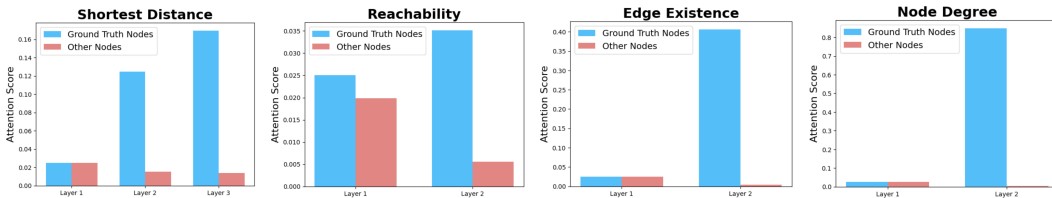

Figure 3: **Layer-by-Layer Attention Interpretation.** This figure compares the mean attention scores for relevant nodes with irrelevant nodes across each layer of the model in 4 graph reasoning tasks. The increased scores in higher layers emphasizes the model's capability to learn larger scale structure information and identify relevant graph nodes effectively.

### 4.2.2 INTRINSIC ATTENTION INTERPRETATION

A key strength of our model design is its inherent interpretability, distinguishing it from existing methods. The local embedding matrix $\boldsymbol{Z}^{(0)}$ in the local block captures 1-hop subgraph information, where each $\boldsymbol{z}_{v_i} \in \boldsymbol{Z}^{(0)}$ represents the 1-hop ego-graph centered around node $v_i$. As the transformer layers progress in the global block, they progressively integrate this localized information to capture broader global structures within the graph. This means that embeddings in the higher layers reflect increasingly comprehensive structural details. The attention weights associated with the task query node in the global block provide a direct interpretation of the contribution of each node's structural information to the final prediction, effectively acting as importance scores for each node. This allows for a direct, interpretable insight into how the model makes its decisions.

To illustrate this, we analyze the mean attention scores across all layers, as shown in Figure 3. As we move to higher layers, the attention scores for ground truth nodes increase, while scores for other nodes decrease. This pattern directly confirms that our model effectively focuses on the most important nodes, demonstrating its ability to capture larger-scale structural information. These attention-based interpretations offer clear insights into the model's decision-making process without requiring additional explanation techniques.

### 4.3 GRAPH LEARNING ABILITY ON REAL-WORLD DATASETS.

**Datasets.** To answer the RQ3 and RQ4, we curated seven graph datasets widely recognized in the graph learning community, varying in scale, domains, and task types. We adopt Arxiv (Hu et al., 2020b), Cora (Bojchevski & Günnemann, 2018), and Pubmed (Sen et al., 2008) for node-level tasks; Pubmed, WN18RR (Bordes et al., 2013), and FB15k-237 (Bordes et al., 2013) for link-level tasks; and molhiv (Hu et al., 2020a) for graph-level tasks. More dataset details are discussed in Table 7.

**Baselines.** We compare with traditional GNNs including GCN (Kipf & Welling, 2017), GAT (Veličković et al., 2017), GIN (Xu et al., 2018) and GraphSage (Hamilton et al., 2017). For graph transformer-based baseline we include GTN (Yun et al., 2019) and Graphormer (Ying et al., 2021). For LLMs, we compare Zero-shot and Few-shot performance using GPT 3.5 (Ye et al., 2023a) for node-level tasks, and Llama-2-7B finetuned InstructGLM (Ye et al., 2023b) for both node and link-level tasks. For the graph-level task, we compare with a GNN-LLM hybrid model Momu (Su et al., 2022) for molecular graphs. Note that we use Mamba (Gu & Dao, 2023) as a baseline for graph-level task as no Transformer-based LLM is computationally feasible for training on real-world graph-level tasks. OFA (Liu et al., 2024a), a hybrid GNN-LLM model, is also selected as a baseline due to its strong performance on link-level tasks.

**Quantitative Results.** As demonstrated in Tables 3, 4, and 5, our method consistently delivers competitive performance across node, link, and graph-level tasks. Compared to traditional GNNs, our model surpasses their performance for both node and link-level tasks with large margins. In comparison to hybrid GNN-LLM methods, our model notably outperforms the recently developed LLM-equipped OFA, on link-level tasks where OFA is considered especially strong. Furthermore, our model consistently perform favorably against LLM instruction tuning approach - InstructGLM across link level tasks. Although graph transformers perform slightly better in the graph-level task because of their specialized encodings for graph-level tasks, our model produces much higher performance

Table 3: **Node-level comparisons.** This table compares our method with 7 baselines on node-level tasks. Types of methods are grouped based on their underlying approaches. All results are reported as averaged Accuracy with standard deviations across 3 random runs. The best and second-best results are highlighted in **bold** and underline respectively.

| | GNN | | | GT | LLM-inference | | LLM-finetuning | |
|---|---|---|---|---|---|---|---|---|
| Dataset | GCN | GAT | GraphSage | Graphormer | Zero-shot | Few-shot | InstructGLM | HLM-G (Ours) |
| arxiv | $71.74_{\pm 0.29}$ | $73.65_{\pm 0.11}$ | $71.49_{\pm 0.27}$ | $72.81_{\pm 0.23}$ | $74.0_{\pm 0.00}$ | $72.9_{\pm 0.00}$ | $\mathbf{75.70_{\pm 0.12}}$ | $\underline{74.81_{\pm 0.07}}$ |
| Pubmed | $88.9_{\pm 0.32}$ | $83.28_{\pm 0.12}$ | $86.85_{\pm 0.11}$ | $88.24_{\pm 1.50}$ | $88.6_{\pm 0.00}$ | $85.0_{\pm 0.00}$ | $\underline{93.84_{\pm 0.25}}$ | $\mathbf{94.62_{\pm 0.13}}$ |
| Cora | $87.78_{\pm 0.96}$ | $76.70_{\pm 0.42}$ | $86.58_{\pm 0.26}$ | $80.41_{\pm 0.30}$ | $66.1_{\pm 0.00}$ | $65.1_{\pm 0.00}$ | $\underline{87.08_{\pm 0.32}}$ | $\mathbf{88.5_{\pm 0.43}}$ |

Table 4: **Link-level comparisons.** This table demonstrates the comparisons between our method and 4 baselines on link-level tasks. We evaluate Pubmed by ROC-AUC, others by Accuracy.

| | GNN | | GNN-LLM | LLM-finetuning | |
|---|---|---|---|---|---|
| Dataset | GCN | GIN | OFA | InstructGLM | HLM-G (Ours) |
| Pubmed | $91.10_{\pm 0.50}$ | $67.88_{\pm 5.45}$ | $\underline{98.21_{\pm 0.02}}$ | $95.92_{\pm 1.91}$ | $\mathbf{98.47_{\pm 0.18}}$ |
| FB15k-237 | $74.20_{\pm 1.10}$ | $70.70_{\pm 1.80}$ | $\underline{95.54_{\pm 0.06}}$ | $64.39_{\pm 0.98}$ | $\mathbf{95.71_{\pm 0.13}}$ |
| WN18RR | $67.40_{\pm 2.40}$ | $57.30_{\pm 3.40}$ | $\underline{96.91_{\pm 0.11}}$ | $63.8_{\pm 1.5}$ | $\mathbf{98.09_{\pm 0.54}}$ |

Table 5: **Graph-level comparisons.** This table demonstrates the comparisons between our method and 6 baselines on graph-level task. We evaluate molhiv by ROC-AUC.

| | GNN | | | GT | GNN-LLM | LLM-finetuning | |
|---|---|---|---|---|---|---|---|
| Dataset | GCN | GAT | GIN | GTN | Momu | Mamba | HLM-G (Ours) |
| molhiv | $75.49_{\pm 1.63}$ | $74.45_{\pm 1.53}$ | $76.26_{\pm 1.41}$ | $\mathbf{77.67_{\pm 1.49}}$ | $75.92_{\pm 0.85}$ | $74.23_{\pm 0.12}$ | $\underline{76.49_{\pm 0.33}}$ |

than them in node-level tasks. It is noteworthy that while LLM-only models excel in node-level tasks, they experience a marked decline in performance on link-level tasks, validated their limitations in processing structural information mentioned in Section 2. A crucial factor in our model's adaptability across all task levels is our pooling parameter $\alpha$ discussed in detail in Appendix F.1. This enables our model to adjust its reliance on structural or feature-based information, thereby allowing it to generalize well across all levels. Our model's ability to dynamically adjust $\alpha$ provides a significant advantage, making it more versatile and capable of handling a wide range of graph-centric tasks.

Overall, our experiments highlight our model's computational efficiency (Appendix D.3), ability to process structural information, interpretability, and effectiveness across diverse tasks. For additional experiments and ablation studies, please refer to Appendix D and Appendix F respectively.

## 5 CONCLUSIONS AND DISCUSSIONS

In this paper, we introduce a novel Language Model Design to tackle the complexities of non-Euclidean structures commonly found in graphs. While language models excel in text-centric applications, they often struggle with the intricate structures of graph data, leading to significant information loss and computational challenges. Additionally, the context length, which involves the natural language description of a graph, can become enormous for real-world datasets, rendering them ineffective for graphs. Our method sets itself apart by designing a hierarchical architecture to process the graph structure and enhance computational efficiency and interpretability. We show that our model yields promising results in graph reasoning tasks as well as robust and consistent performance on real-world datasets, outperforming most models designed for similar purposes.

This work paves the way for future research in language models for graph learning, establishing a solid foundation for innovation and providing valuable insights into this emerging field. Our findings significantly narrow the gap between conventional language models and graph data, expanding the potential applications and improving the effectiveness of language models in handling structured data. We hope this work can shed light on the future direction of LLM-based graph learning.

## REPRODUCIBILITY STATEMENT

To ensure the reproducibility of this work, we provide full details for the experiments including all the datasets used, training setup, architecture and hardware used in Appendix C. We also provide an anonymous code link containing the implementation of our method: `https://anonymous.4open.science/r/HLM_G/`.

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

# Appendix of HLM-G

CONTENTS

## A    BROADER IMPACTS

Our research aims to enhance the understanding of graph structures through language models (LMs), marking a modest but significant step toward improved graph reasoning capabilities. This foundational effort seeks to refine how LMs interpret complex graph data, aspiring to inspire further research in this domain. Given the exploratory nature of our work, we have not identified specific negative societal impacts or potential for malicious use directly attributable to our research. Nevertheless, we recognize that all technological advancements carry inherent risks.

In alignment with responsible research practices, we suggest continuous monitoring of developments in the application of LMs to graph data analysis. As these models evolve to handle more complex tasks, maintaining vigilance becomes crucial to preemptively address any emerging risks before they manifest. Our commitment to ethical conduct underpins our research methodology, which is designed to avoid harm and does not involve human subjects, thus mitigating potential ethical concerns related to privacy and fairness. By promoting ongoing assessment and adopting a proactive approach to research governance, we aim to ensure that our contributions positively impact the field and adhere to the highest standards of ethical research.

## B    RELATED WORKS

**Graph Neural Networks (GNNs).**  Graph Neural Networks (GNNs) have emerged as a powerful framework for learning over graph-structured data (Kipf & Welling, 2017; Gilmer et al., 2017; Veličković et al., 2018; Wu et al., 2020; Liu et al., 2020). GNNs operate by iteratively aggregating information from a node's neighbors, thereby learning node representations that capture the local structure and features of the graph. This message-passing mechanism enables GNNs to be highly effective in tasks such as node classification, link prediction, and graph classification. However, despite their success, GNNs are often challenged by issues such as over-smoothing in deeper networks (Rusch et al., 2023) and difficulties in handling long-range dependencies (Sanford et al., 2024), which can limit their effectiveness on larger and more complex graphs.

**Graph Transformer (GT).**  Graph Transformers (GTs) (Yun et al., 2019; Rampášek et al., 2022) represent a more recent approach that aims to capture global dependencies within graph data using self-attention mechanisms. Inspired by the success of transformers in NLP tasks, GTs adapt the self-attention mechanism to graph-structured data, allowing them to capture both local and global interactions simultaneously. This approach helps address some of the limitations of GNNs in learning long-range dependencies. However, Graph Transformers often require additional architectural complexities (Black et al., 2024), such as centrality encoding, edge features, and spatial encodings, to effectively represent graph structures. These added complexities can lead to increased computational demands and make them less interpretable compared to conventional GNNs.

**Transformer Block in Language Models.** In a transformer model, each block processes an input sequence $H_i = \{h_1, h_2, \ldots, h_{n_i}\}$ to output an updated sequence $H_{i+1}$. A transformer block is structured around an attention mechanism and a feedforward network, both supplemented by residual connections and layer normalization. The multi-head attention mechanism processes the sequence $H_i$, formulated as $\text{Attention}(Q, K, V) = \text{softmax}\left(\frac{QK^T}{\sqrt{d_k}}\right) V$ where $Q, K, V$ are queries, keys, and values derived from $H_i$, and $d_k$ is the dimension of keys. This output is then combined with the original input $H_i$ and normalized: $\text{Output}_{\text{attention}} = \text{LayerNorm}(H_i + \text{Attention}(H_i))$. Following this, a position-wise feedforward network processes each position in $\text{Output}_{\text{attention}}$, described by $\text{FFN}(x) = \max(0, xW_1 + b_1)W_2 + b_2$ with $W_1, W_2, b_1, b_2$ as the network parameters. The final output $H_{i+1}$ for the block is computed by applying another layer normalization on the summation of the feedforward network output and the attention output: $H_{i+1} = \text{LayerNorm}(\text{Output}_{\text{attention}} + \text{FFN}(\text{Output}_{\text{attention}}))$. This architecture allows the transformer to capture and process dependencies across the input sequence, enabling deep contextual understanding that propagates through successive layers of the model.

**Comparisons to Prior Work.**  LLM-only methods commonly fail to effectively learn from graph data due to computational feasibility and the loss of graph structural information. In contrast, our model addresses these challenges with a local-to-global hierarchical design that efficiently leverages graph structure. Hybrid GNN-LM approaches typically encounter problems with task-specific designs

and limited interpretability. In comparison, our method is inherently task-agnostic and demonstrates high interpretability. When compared to closely related conventional Graph Transformers, which necessitate complex designs for centrality, edge, and spatial encoding, our method streamlines the process by exclusively using natural language input, eliminating the need for these elaborate encodings.

## C  EXPERIMENT DETAILS

### C.1  DETAILS ABOUT THE DATASETS

In this section, we describe in detail the datasets used for our experiments. We first describe the Graph Reasoning dataset followed by the real world datasets.

#### C.1.1  GRAPH REASONING

Several works have proposed benchmarks for graph reasoning, such as the NLGraph (Wang et al., 2024a) and GraphQA (Fatemi et al., 2023). However, upon closer examination, we observed that these benchmarks suffer from significant class imbalance, with some classes having far more data points than others. For example, in the cycle dataset of GraphQA, 82% of the data samples contain at least one cycle. Some works like Graphtoken (Perozzi et al., 2024) have leveraged this dataset, with their proposed architecture achieving 83% accuracy on the cycle dataset. This raises concerns about whether the models are truly reasoning on the datasets or simply making majority label predictions. Additionally, the majority of graphs in these benchmarks have a small number of nodes, typically ranging from 5 to 20. In reality, we expect real-world graph datasets to be much larger than this.

To address these issues, we propose a new benchmark constructed using a random graph generator. Importantly, all datasets in our benchmark are balanced, enabling us to evaluate the true graph reasoning ability of language models accurately. Training and validation graphs contain up to 40 nodes with test set containing exactly 40 nodes. .

In this section we describe our random graph generator used for creating graph reasoning datasets.

**Pre-defined graphs** To ensure that our generator is well covered, we first include common graphs including Cyclic graphs, Star graphs, Complete graphs, Path graphs, Tree graphs, Wheel graphs and Barbell graphs. All of these graphs can be created using NetworkX documentation[1].

**Random graphs** A graphon is a function $W : [0,1]^2 \to [0,1]$ that takes 2 values $v_1$, $v_2 \in [0,1]$ for each pair of nodes and returns the probability $p \in [0,1]$ for an edge between these 2 nodes. The function $W$ can be any function that takes 2 values $v_1, v_2 \in [0,1]$ and returns $p \in [0,1]$. Given two values $v_1$ and $v_2$, we implement following functions:

1. Constant graphon: Returns a random number $p \in [0.3, 0.7]$

2. Sparse graphon: Returns a small random number $p \in [0.05, 0.15]$

3. Dense graphon: Returns a big random number $p \in [0.8, 1.0]$

4. Linear graphon: Returns $p = v_1 * v_2$

5. Quadratic graphon: Returns $p = v_1^2 * v_2^2$

6. Sigmoidal graphon: Returns $p = \frac{1}{1+\exp(-10(u-v))}$

7. Step graphon: Returns $p = 1$ if $v_1 \geq t$ and $v_2 \geq t$ for some random threshold $t \in [0,1]$

8. Sin graphon: Returns $p = \sin(\pi v_1) \cdot \sin(\pi v_2)$

9. Avg graphon: Returns $p = (v_1 + v_2)/2$

10. Exp. decay graphon: Returns $p = \exp(-(v_1^2 + v_2^2))$

11. Softmax graphon: Returns $p = \frac{\exp(v_1)}{\exp(v_1)+\exp(v_2)}$

---

[1] https://networkx.org/documentation/stable/index.html

The value $v_i$ for the $i^{\text{th}}$ node is randomly initialized for each node. Using these formulations, we prepare our benchmark for structural reasoning tasks. For every task, we extract a graph from our Random Graph Generator and assign it a label depending on the task. We collect equal number of graphs for every label to prevent bias towards majority class.

Table 6: Summary of Graph Analysis Tasks and Their Dataset Specifications

|  | Distance | Cycle Detection | Edge Count | Reachability | Edge Existence | Connected Components | Node Degree |
|---|---|---|---|---|---|---|---|
| **#Classes** | 6 | 2 | 70 | 2 | 2 | 38 | 39 |
| **Dataset Size Used** | 20000 | 4000 | 14000 | 4000 | 4000 | 19000 | 8000 |

Armed with the general-purpose graph dataset generator, we adapt synthetically generated graphs to various graph reasoning tasks with varying complexities and describe the problem setups in a structured manner. Specifically, we first generate subsets of base graphs for each task by controlling node quantity and graph density. We then tailor these base graphs for specific tasks and design queries to assess the models' capabilities accordingly. These 7 datasets summarized in Table 6 are detailed below. A random split of 80/10/10 is used for training , validation and test sets.

- **Task 1: Shortest Distance**
  Given a graph $G = \{V, E\}$, predict the shortest distance between two nodes $v_i$ and $v_j$, categorized into six classes from 0 to 5. Class 0 indicates no path exists, while classes 1 to 5 represent distances from 1 to 5 edges. The query posed is: "What is the shortest distance between nodes $v_i$ and $v_j$?"

- **Task 2: Cycle Detection**
  In a graph $G = \{V, E\}$, determine if a cycle exists. The task classifies graphs into two categories: presence or absence of a cycle. The question asked is: "Is the graph cyclic?"

- **Task 3: Edge Count**
  Our random graph generator can produce graphs with over 700 edges. To minimize the required training size, we categorize sets of 10 edges into a single class. Specifically, graphs with 1 to 10 edges are classified as class 0, 11 to 20 as class 1, continuing in this manner up to 691 to 700, which are classified as class 80. In Section 6, we explore a similar generation task that features 700 distinct classes (with no grouping of graphs), and it demonstrates comparable performance.

- **Task 4: Reachable**
  In a graph $G = \{V, E\}$, predict whether there is a reachable path between two nodes $v_i$ and $v_j$. The query for this task is: "Are nodes $v_i$ and $v_j$ reachable from each other?"

- **Task 5: Edge Existence**
  Determine if an edge exists between two nodes in a graph, represented as $G = \{V, E\}$. The posed query is: "Does an edge exist between nodes $v_i$ and $v_j$?"

- **Task 6: Connected Components**
  Predict the number of connected components in a graph $G = \{V, E\}$. A component is a set of nodes that are reachable from one another. Specifically if $v_i \in C_1$ and $v_j \in C_2$ where $C_1$ and $C_2$ are different components then there exists no path from $v_1$ to $v_2$. The query is: "How many connected components does the graph have?"

- **Task 7: Node Degree**
  Estimate the degree of a node in the graph, representing the number of direct connections or neighbors the node has. The question is: "What is the degree of node $v_i$?"

### C.1.2 REAL WORLD DATASETS

We conduct experiments on 7 different Text Attributed Graph (TAG) datasets. All of these graphs have node features available in natural language. The datasets are concisely summarized in the table below, followed by detailed descriptions of each dataset.

**Cora** dataset, sourced from the GitHub repository as described in Chen et al. (2024c) , is a citation network in the computer science domain. Each node in this dataset represents a research paper, with raw text features consisting of the paper's title and abstract. The edges between nodes indicate citation relationships. Nodes are labeled according to the category of the paper, encompassing

Table 7: Datasets summary for real world graphs with text attributes

| Dataset | Domain | Task | # Graphs | Avg. #Nodes | Avg. #Edges | # Classes |
|---------|--------|------|----------|-------------|-------------|-----------|
| Cora | Citation | Node | 1 | 2,708 | 10,556 | 7 |
| ogbn-arxiv | Citation | Node | 1 | 169,343 | 1,166,243 | 40 |
| PubMed | Citation | Node | 1 | 19,717 | 44,338 | 3 |
| PubMed | Citation | Link | 1 | 19,717 | 44,338 | 2 |
| FB15k-237 | Knowledge | Link | 1 | 14,541 | 310,116 | 237 |
| WN18RR | Knowledge | Link | 1 | 40,943 | 93,003 | 11 |
| HIV | Molecule | Graph | 41,127 | 25.5 | 27.5 | 2 |

seven possible classes. For our study, we focus on node-level prediction, specifically predicting the category of each paper based on its features and structure. We use the 60-20-20 random split for training, validation and testing.

**PubMed** dataset comprises 19,717 scientific publications from the PubMed database, specifically related to diabetes, categorized into one of three classes. The citation network includes 44,338 links. Each node represents a research paper, with raw text features including the paper's title and abstract. Our study involves both node classification and link classification tasks on the PubMed dataset. The raw text data of PubMed dataset was collected from GitHub repository provided in Chen et al. (2024c).

For node classification, we use a 60-20-20 random split for training, validation, and testing. For link classification, the goal is to predict whether two nodes are directly connected. Following the methodology of OFA (Liu et al., 2024a), we use an 85-5-10 random split. In the link classification task, The training, validation and testing set is created using existence edges as positive samples and an equal number of negative samples by checking for the absence of an edge between nodes. The evaluation metric for the link-level task is the AUC.

**ogbn-arXiv**[2] is a directed graph representing the citation network among Computer Science (CS) arXiv papers. The task involves predicting the 40 subject areas of these papers, such as cs.AI, cs.LG, and cs.OS, which are manually labeled by the authors and arXiv moderators. We follow the standard split for this dataset: training on papers published until 2017, validating on those from 2018, and testing on papers published since 2019. The raw text data of the ogbn-arxiv was collected using the same protocol as the GitHub repository provided in Prodigy (Huang et al., 2024).

**Molhiv**[3] dataset is a molecular property prediction dataset adopted from the MoleculeNet (Wu et al., 2018). The dataset contains 41127 molecules each represented as a graph with atom as nodes and bonds as edges. Each atom has 9 discrete features. 1 of the features (Chirality) is common for all atoms and is therefore not considered. The rest of the features are: Atomic Number, Degree of atom, Formal charge, Number of connected Hydrogen, Radical electrons, Hybridization, Aromaticity and Ring. These features can be converted to natural language using only a few lines of code. Similarly the bonds between any 2 atoms can be of 4 types: single, double, triple or aromatic. Each of these bonds also has a boolean property: conjugated. Therefore any edge can be represented using the bond type and whether or not it is conjugated.

Here we perform graph level classification where objective is to classify a molecule as HIV inhibitor or not. The metric used here is AUC.

**WN18RR** is a link prediction dataset created from WN18, which is a subset of WordNet. WN18RR dataset contains 93,003 triples with 40,943 entities and 11 relation types. Here we perform link classification where we classify any edge in 11 possible edge types. This dataset is extracted from

---

[2]ogbn-arXiv is released under license ODC-BY.

[3]ogbg-molhiv is released under license MIT.

GitHub repository[4].

**FB15k-237** is a knowledge graph that contains knowledge base relation triples and textual mentions of Freebase entity pairs. It contains 310,116 triples with 14,541 entities and 237 relation types. Here we perform link classification where we classify any edge in 237 possible edge types. The raw text data of nodes in FB15K237 was collected from the same Github repository as WN18RR.

## C.2 TRAINING AND OPTIMIZATION SETTINGS

For the graph reasoning datasets, we train our model from scratch, with the input being the natural language description of the graph structure. In the local block, we employ a BERT-like architecture utilizing a special intra-node masking scheme that masks out language tokens belonging to different nodes. Across all reasoning datasets, we use 4 local block layers. For the global block, we utilize 2 layers for most datasets, except for the Shortest Distance, Edge Count and Number of Connected Components datasets, where 3 global block layers are used. Our observations indicate that more complex tasks benefit from an increased number of global block layers, which enhances overall performance.

We adopt the Adam optimizer (Kingma & Ba, 2014) throughout the training phase, with a learning rate of $5e^{-6}$, weight decay of 0.1, $\beta_1 = 0.9$, and $\beta_2 = 0.95$. Across all datasets, the training consists of 5 epochs, with a batch size of 16 for graph reasoning datasets and 8 for real-world datasets. The shared parameters for all tasks and datasets used in our language model $M_L(G)$ are summarized in Table 8.

Table 8: Parameters used for the language model.

| Parameter | Value |
| --- | --- |
| Activation | gelu |
| Attention Dropout | 0.1 |
| Dimension | 768 |
| Dropout | 0.1 |
| Hidden Dimension | 3072 |
| Max Position Embeddings | 4096 |
| Number of Heads | 12 |
| Number of Local Block Layers | 6 |

We attempted to leverage pretrained models such as BERT (Devlin et al., 2018), SBERT (Reimers & Gurevych, 2019), DistilBERT (Sanh, 2019), and Llama 2 7B (Touvron et al., 2023) as the lower block, but found no performance gains on graph reasoning tasks; in fact, performance declined when these models were not fine-tuned. This suggests that these pretrained models do not acquire graph structure-related information during pretraining. Our experiments indicate that fine-tuning just 4 layers of the lower block is sufficient to achieve state-of-the-art performance on graph reasoning tasks.

For real-world datasets (Tables 3, 4, and 5), we employ DistilBERT in the local block and fine-tune it. Given that these datasets contain textual node and edge features, pretrained models are better equipped to understand these features. The number of higher block layers for each dataset is set as follows: 4 for Cora, Pubmed, WN18RR, and FB15k-237, 2 for molhiv, and 6 for Arxiv.

We observed that using larger models yields improved performance on node-level tasks, as depicted in Table 9. This is expected since node features play a more critical role in making accurate node-level predictions within real-world datasets and these larger models are better equiped to understand these text based features.

---

[4]https://github.com/villmow/datasets_knowledge_embedding/tree/master

Table 9: Performance comparison with different text encoders for Cora and Pubmed.

|  | **DistilBERT** | **SBERT** | **Llama-2** |
|---|---|---|---|
| Cora | 87.9% | 88.9% | **89.2%** |
| Pubmed | 94.1% | 93.9% | **94.9%** |

## C.3 SOFTWARE AND HARDWARE

Our implementation is under the architecture of PyTorch (Paszke et al., 2019) and PyG (Fey & Lenssen, 2019). The deployment environments are Ubuntu 18.04 with 48 Intel(R) Xeon(R) Silver 4214R CPU @ 2.40GHz, 755GB Memory, and graphics cards NVIDIA RTX A6000.

## D ADDITIONAL EXPERIMENT RESULTS

### D.1 DOWNSTREAM TASK PERFORMANCE

To assess the adaptability and transferability of our proposed model across different graph domains and task levels, we evaluated its performance on downstream tasks. Specifically, we examined how well the model, when trained on one task level (e.g., node, link, or graph), could adapt to perform effectively on another.

**Experimental Setup:** We pretrained our model on three distinct datasets representing different task levels: Arxiv (Node-level), Molhiv (Graph-level), and Pubmed (Link-level). Each pretrained model was then fine-tuned on a variety of downstream tasks by updating only the final classification layer for 5 epochs with a learning rate of $4e^{-5}$. This setup allowed us to evaluate the model's ability to leverage learned knowledge and adapt to completely different downstream tasks.

**Results and Analysis:** The results in Table 10 demonstrate the impressive transferability of our model. For example, the model pretrained on the Arxiv (Node-level) dataset achieved an 87.8% accuracy on the PubMed Link task, outperforming the performance of fully trained GIN despite being trained exclusively on node-level information initially. Similarly, the model pretrained on the Molhiv (Graph-level) dataset delivered competitive results on both node-level (Cora) and link-level (PubMed) tasks, showcasing its ability to adapt to diverse task requirements.

These insights highlight the versatility of our approach, indicating that our model can effectively generalize knowledge from one graph domain to another. Our language model design not only captures graph structures efficiently but can also be fine-tuned for a wide range of downstream applications with limited training, making it a valuable asset for practical real-world applications.

Table 10: Downstream task performance with different pretraining datasets. The model's performance was evaluated after fine-tuning only the classification layer for 5 epochs.

| Pretrained \Downstream | Cora (Node) | Pubmed (Node) | Pubmed (Link) | Molhiv (Graph) |
|---|---|---|---|---|
| Arxiv (Node) | 80.6 | 83.8 | **87.8** | 72.2 |
| molhiv (Graph) | 73.9 | 75.4 | 86.6 | - |
| Pubmed (Link) | 71.6 | 77.5 | - | 72.5 |

### D.2 GENERATION TASKS

The current architecture employs local and global transformer blocks and a classification layer for final prediction. For generation on graphs, we need a Decoder model that can generate the output. For this, we take inspiration from GraphLLM (Chai et al., 2023) and leverage Prefix-Tuning (Li & Liang, 2021) for fine-tuning a Frozen Decoder LLM with HLM-G encoder.

**Prefix Tuning** Given a pre-trained LLM with an $L$-layer transformer, prefix tuning prepends $K$ trainable continuous tokens (prefixes) to the keys and values of the attention at every transformer layer. Taking the $l$-th attention layer as an example ($l < L$), prefix vectors $\boldsymbol{P}_l \in \mathbb{R}^{K \times d^{\mathrm{M}}}$ is concatenated

with the original keys $\boldsymbol{K}_l \in \mathbb{R}^{* \times d^{\mathrm{M}}}$ and values $\boldsymbol{V}_l \in \mathbb{R}^{* \times d^{\mathrm{M}}}$, where $d^{\mathrm{M}}$ is the dimension of LLM, formulated as:

$$\boldsymbol{K}_l' = [\boldsymbol{P}_l; \boldsymbol{K}_l];\ \boldsymbol{V}_l' = [\boldsymbol{P}_l; \boldsymbol{V}_l] \in \mathbb{R}^{(K+*) \times d^{\mathrm{M}}}$$

The new prefixed keys $\boldsymbol{K}_l'$ and values $\boldsymbol{V}_l'$ are then subjected to the $l$-th attention layer of LLM. For simplicity, we denote the vanilla attention computation as $\boldsymbol{O}_l = \mathtt{Attn}(\boldsymbol{Q}_l, \boldsymbol{K}_l, \boldsymbol{V}_l)$. The computation of attention becomes:

$$\boldsymbol{O}_l = \mathtt{Attn}(\boldsymbol{Q}_l, [\boldsymbol{P}_l; \boldsymbol{K}_l], [\boldsymbol{P}_l; \boldsymbol{V}_l])$$

We introduce three distinct datasets tailored for graph generation tasks, each with unique complexities and requirements. These tasks are designed to evaluate the model's ability to generate graph structures and properties accurately.

- **Task 1: Shortest Path**
  The objective of this task is to generate the shortest path between two specified nodes in a graph. Given a graph $G$, the query $Q_G$ is formulated as: "What is the shortest path from node $i$ to $j$?", where $i$ and $j$ are valid nodes within $G$. The output is considered correct only if the path generated is both valid and the shortest possible.

- **Task 2: Bipartite Detection**
  This task aims to determine whether a given graph is bipartite. A graph is defined as bipartite if it contains no odd cycles. The challenge for the model is to predict if the graph is bipartite or, if not, to generate an odd cycle. The query $Q_G$ is: "Is the graph bipartite?". An output is deemed correct if it accurately predicts whether the graph is bipartite or identifies an odd cycle when the graph is not bipartite.

- **Task 3: Edge Count**
  This dataset involves predicting the exact number of edges in a graph, enhancing the edge count task detailed in Section 4. Unlike the previous version, this task does not classify edges into pooled groups but requires an exact count. Additionally, the training set does not include all edge counts present in the test set, introducing unseen scenarios. The query $Q_G$ is: "What are the number of edges in the graph?". Correctness is strictly judged on the model's ability to match the exact number of edges in $G$.

Table 11: Performance comparison for zero shot and HLM-G encoder on different generation tasks. Llama-2 7B is used as a decoder in both settings. (across 1 random run).

|  | Shortest Path | Bipartite Detection | Edge Count |
| --- | --- | --- | --- |
| **Zero shot** | 5.2% | 11.7% | 2.1% |
| **HLM-G encoder** | 93.4% | 95.1% | 92.5% |

This data indicates that HLM-G has substantial potential as a powerful graph encoder. The high accuracy across different tasks in our tests demonstrates its effectiveness. Further experiments are necessary to fully explore the zero-shot and few-shot capabilities of HLM-G. These future studies will help validate the model's performance across a broader range of graph-based applications, potentially establishing HLM-G as a useful tool in for leveraging LLMs on graphs.

### D.3 COMPUTATIONAL EFFICIENCY

We systematically compare the training efficiency across various LLM-based methods on graph reasoning datasets and real world dataset.

**Graph Reasoning Datasets.** Our study evaluates multiple fine-tuning approaches, which we categorize into two primary groups: Hybrid GNN-LLM fine-tuning and LLM-only fine-tuning. We present training times for GraphToken (a hybrid method), BERT, Llama 3 (LLM-only), and our proposed HLM-G model (LLM-only fine-tuning). GraphToken utilizes a 4-layer GCN as its GNN encoder with approximately 5.2 million training parameters, resulting in a total parameter count of around 8

billion, comparable to the Lora-trained Llama 3. For BERT, we adopt a 4-layer architecture with four attention heads, yielding 56 million parameters. The parameter count for our HLM-G model varies depending on the dataset, comprising 82 million parameters for tasks such as distance, edge count, and the number of components, and 77 million for reachability, cycle, and edge existence datasets.

Table 12: **Training time and total training time comparison across graph reasoning datasets.** The total training time refers to the duration required to reach the optimal validation checkpoint.

| Dataset | GraphToken | | Llama 3 | | BERT | | HLM-G (ours) | |
|---|---|---|---|---|---|---|---|---|
| | Time/Epoch | Total Time | Time/Epoch | Total Time | Time/Epoch | Total Time | Time/Epoch | Total Time |
| Distance | 30 mins | 20 hours | 17 hours | 34 hours | 2 hours 30 mins | 7.5 hours | 2 hours | 6 hours |
| Reachability | 9 mins | 8 hours | 6 hours | 30 hours | 2 hours | 10 hours | 45 mins | 1 hour 30 mins |
| Cycle | 9 mins | 12 hours | 7 hours | 21 hours | 45 mins | 1 hour 30 mins | 40 mins | 40 mins |
| Edge Count | 33 mins | 36 hours | 12 hours | 48 hours | 3 hours 30 mins | 24.5 hours | 3 hours | 6 hours |
| Edge Existence | 9 mins | 8.5 hours | 6 hours | 24 hours | 45 mins | 1 hour 30 mins | 40 mins | 40 mins |
| Connected Components | 15 mins | 18 hours | 12 hours | 36 hours | 2 hours | 14 hours | 1 hour | 12 hours |
| Node Degree | 17 mins | 12 hours | 11 hours | 33 hours | 1 hour 30 mins | 6 hours | 1 hour | 1 hour |

Table 12 offers a comprehensive comparison of training times among various fine-tuning methods. Despite the HLM-G model having 20 to 30 million more parameters than BERT, its hierarchical dual-block architecture significantly reduces both the training time per epoch and the total time to convergence. In contrast, GraphToken, while achieving shorter training times per epoch, requires a substantially higher number of epochs to reach convergence due to its use of a GCN encoder. Additionally, the training times for Llama 3 are notably high, as expected, due to the model's extensive number of parameters and the maximum input prompt length of 4096, which necessitates longer training durations. In comparison, our HLM-G model exhibits considerable improvements in training efficiency, highlighting the computational advantages of our approach, especially in managing large-scale graph reasoning tasks.

**Real-world Datasets.** We evaluated the training times of our model, HLM-G, against InstructGLM for node and link prediction tasks, as InstructGLM does not support graph-level tasks. For graph-level tasks, we compared HLM-G with Mamba. InstructGLM uses Llama-2 7B as its backbone and incorporates Lora with a rank of 16, resulting in 8.2 million trainable parameters. The trainable parameter count for Mamba is approximately 91.8 million. For HLM-G, the number of trainable parameters varies depending on the number of layers in the higher block (as detailed in Appendix C.2), ranging from 76 million for Molhiv to 86 million for datasets such as Pubmed, Cora, and knowledge graphs, and up to 96 million for Arxiv.

Table 13: **Training time and total training time comparison across real-world datasets.** Total training time refers to the duration required to reach the optimal validation checkpoint.

| Dataset | Mamba | | InstructGLM | | HLM-G (ours) | |
|---|---|---|---|---|---|---|
| | Time/Epoch | Total Time | Time/Epoch | Total Time | Time/Epoch | Total Time |
| Pubmed Node | - | - | 23 hours 10 mins | 69 hours 30 mins | 2 hours 5 mins | 2 hours 5 mins |
| Pubmed Link | - | - | 23 hours 10 mins | 46 hours 20 mins | 10 hours 30 mins | 21 hours |
| Arxiv | - | - | 105 hours | 210 hours | 7 hours | 28 hours |
| WN18RR | - | - | 34 hours | 68 hours | 2 hours 10 mins | 4 hours 20 mins |
| FB15k-237 | - | - | 56 hours | 56 hours | 5 hours | 25 hours |
| Molhiv | 6 hours | 150 hours | - | - | 3 hours | 18 hours |

Table 13 presents a comparison of training times across real-world datasets, demonstrating the computational efficiency of our HLM-G model relative to other fine-tuned language models. The results clearly highlight HLM-G's capability to perform graph-based tasks efficiently while maintaining high performance. Particularly notable is the significant reduction in training time compared to InstructGLM, especially in larger datasets. This efficiency underscores where our model is most useful.

The real-world datasets used in these comparisons are characterized by their immense size and complex descriptions, factors that typically challenge traditional LLMs. Our HLM-G model is specifically designed to excel in these environments. Unlike conventional LLMs, which may struggle with the scale and specificity of graph-based data, HLM-G leverages its hierarchical architecture to process such data more effectively. This design enables HLM-G to handle the intricate details and

vast data volumes more adeptly, making it particularly suited for tasks involving extensive real-world graphs. This advantage makes HLM-G a preferred tool for applications requiring robust and efficient graph reasoning capabilities.

# E    INTERPRETATION

## E.1    INTERPRETATION GROUND TRUTH

In the context of graph reasoning datasets, any graph can be partitioned into two distinct sets of nodes: *citical ground truth nodes*, which are directly responsible for the final prediction, and *non-critical nodes*, which do not influence the prediction either directly or indirectly. Due to the importance of focusing on structurally relevant nodes, we exclude datasets such as components and edge count where each node is integral to the final prediction. This exclusion is crucial as it allows us to experimentally investigate our model's attention mechanisms towards nodes that are truly significant in the reasoning process. Detailed ground truth sets for 3 link-level and 1 node-level task are described below.

**Edge Existence** In the edge existence task between two nodes $u$ and $v$, the nodes $u$ and $v$ themselves are sufficient for determining the presence of an edge, thus forming the ground truth:

$$GT = \{u, v\}$$

**Shortest Distance** For the shortest distance between nodes $u$ and $v$, ground truth nodes include all nodes lying on any shortest path. Let $l$ be the shortest path length, then ground truth is simply union over all these nodes:

$$GT = \bigcup_{i=1}^{m_l} \{u, a_1^i, a_2^i, \ldots, a_{l-1}^i, v\}$$

where $m_l$ is the number of shortest paths.

**Reachability Dataset** Unlike the shortest path dataset, reachability requires consideration of all nodes in all possible paths from $u$ to $v$, including those beyond the shortest path. If $n$ is the total number of nodes in the graph, the ground truth set includes:

$$GT = \bigcup_{j=l}^{n-1} \bigcup_{i=1}^{m_j} \{u, a_1^i, a_2^i, \ldots, a_{j-1}^i, v\}$$

where $m_l$ is the number of paths of length $j$, $j \in \{l, l+1...n-1\}$. This represents a more holistic understanding of the graph's connectivity by including paths of length $l$ through $n-1$.

**Node Degree** For node degree tasks focused on a single node $u$, the determination of degree relies solely on its direct connections to other nodes in the graph.. The ground truth is straightforward in this case:

$$GT = \{u\}$$

Together, these definitions facilitate a comprehensive evaluation of our model's capability to handle various structural reasoning tasks, each necessitating a specific set of nodes as ground truth based on task requirements.

## E.2    EXPLANATION AS INTERPRETATION

It is challenging to compare interpretability performance with methods that are not interpretable or have different interpretation formats. To achieve such comparisons, we propose to use explanations of models as interpretations. [5] However, explanations provided by explainers face their possible performance issue that the produced explanations might not be faithful to the deep model behaviors.

This faithfulness issue requires us to first discover the most faithful explanations for the models before using them as model interpretations. Therefore, instead of using one explainer, we adopt four explainers to select the best explanation for each model on each dataset including Saliency (Simonyan

---

[5]Note that explainers provide post explanations that can be applied to any models, while interpretations are generally produced by the model's specific design, a.k.a., self-interpretable model.

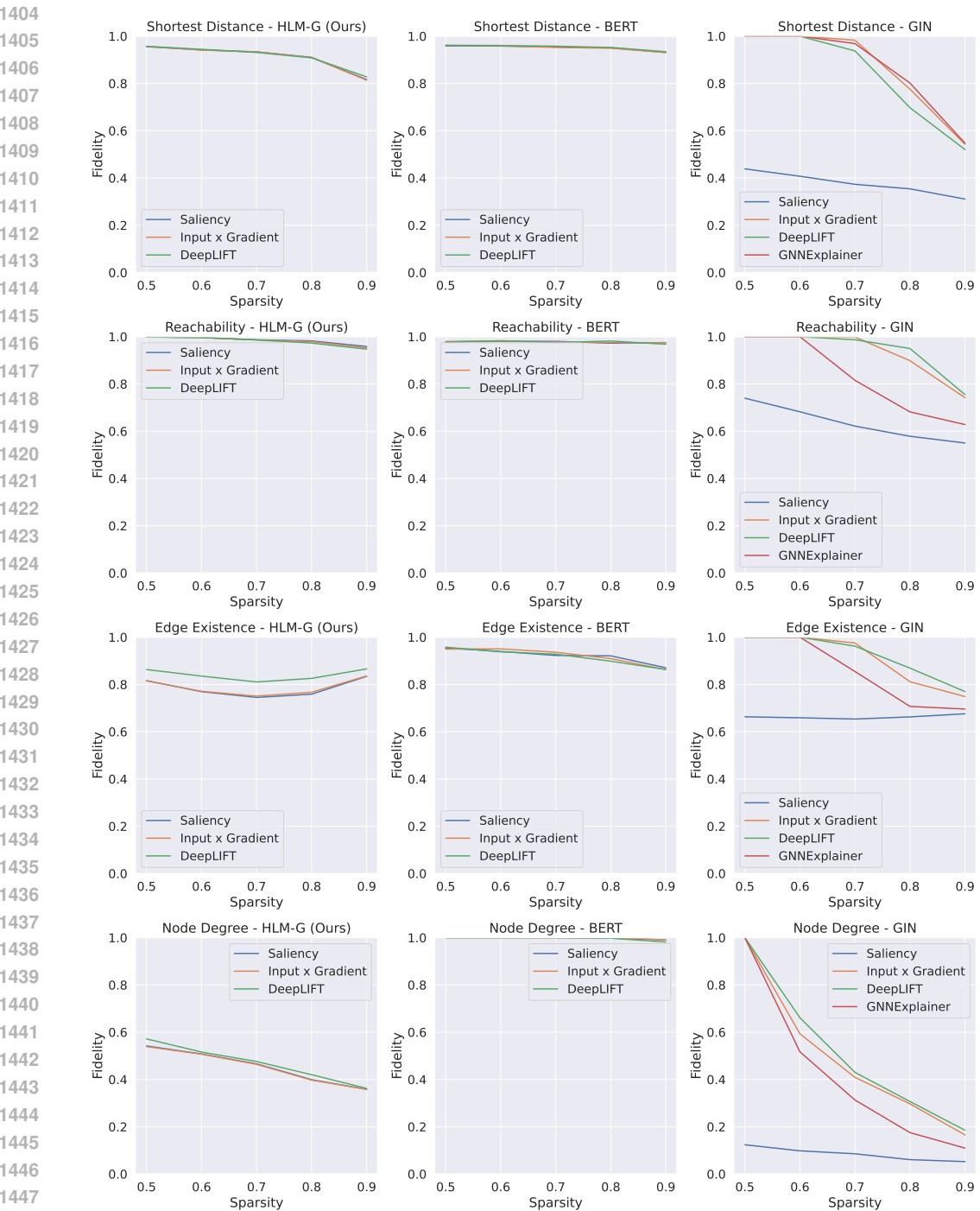

Figure 4: **Fidelity results.** This figure measures the faithfulness of 4 explainers to 3 models using Fidelity scores across different Sparsities. Results should be compared across different explainers within the same dataset and method.

et al., 2013), Input × Gradient (Shrikumar et al., 2016), DeepLIFT (Shrikumar et al., 2017), and GNNExplainer (Ying et al., 2019), where GNNExplainer can be only applied to GNNs. Specifically, we adopt Fidelity- (Yuan et al., 2023), a.k.a., sufficiency Fidelity (Gui et al., 2023), to measure whether an explanation provided by an explainer is faithful to the model behavior. Formally, given $N$ samples, Fidelity can be written as

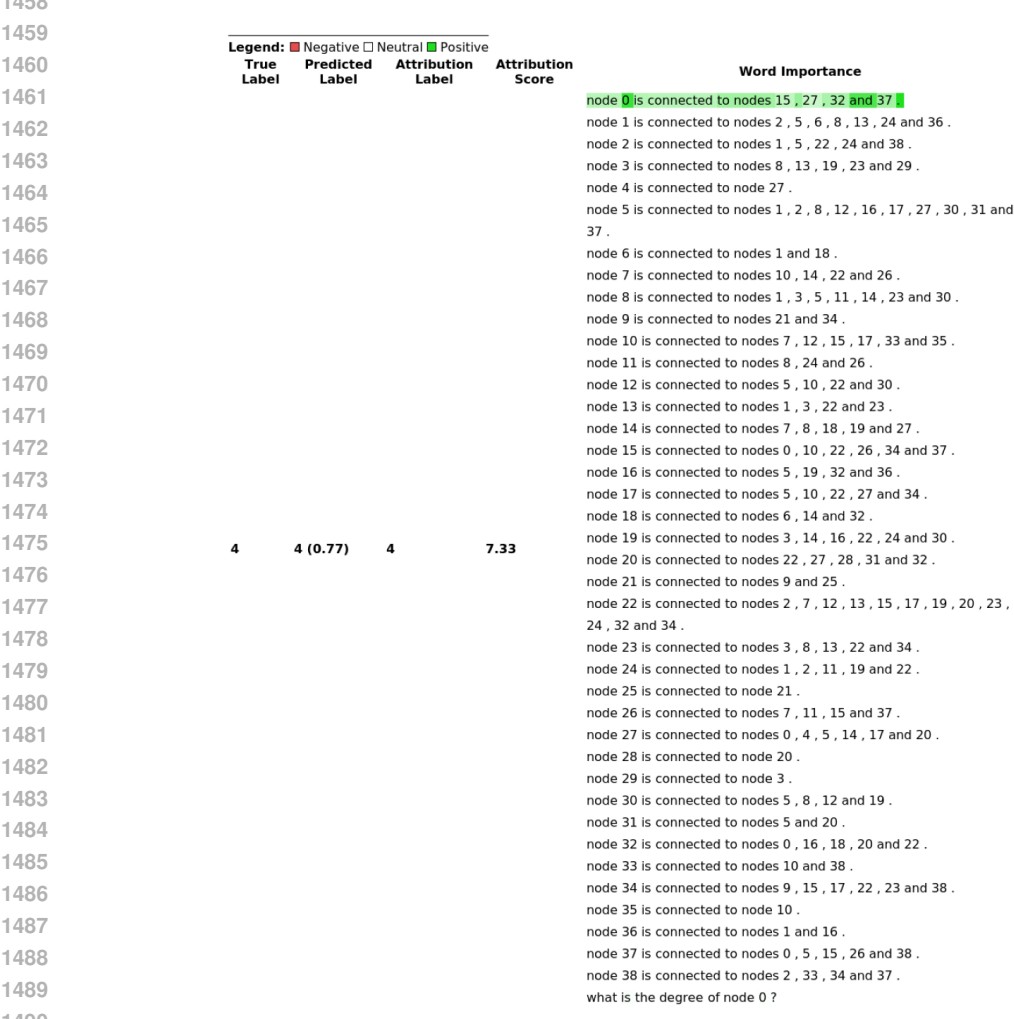

Figure 5: **Interpretation visualization of HLM-G (ours) on the node degree dataset.**

$$\text{Fidelity} = \frac{1}{N} \sum_{i=1}^{N} \left( \mathbb{1}(\hat{y}_i = y_i) - \mathbb{1}(\hat{y}_i^{r_i^k} = y_i) \right), \tag{1}$$

where the sample index $i$ is used as subscription; $\mathbb{1}(\cdot) = 1$ when the given condition is satisfied, otherwise, 0; $\hat{y}_i^{r_i^k}$ indicates the sample $i$'s prediction result using only the top-$k$ important nodes.

Since high Fidelity indicates that the explanation directly reflects the model behavior, the explanation can be used as the model behavior representative. In the experiment, for each dataset and each method, we select the explanation with the highest average Fidelity from all explainers. The Fidelity results are plotted in Figure. 4, where Sparsity denotes the ratio $1 - k/n$; thus, higher Sparsity indicates less important nodes are used.

It is crucial to note that Fidelity results do not reflect the interpretability performance of models, they only show the relation between the explainer and the model and are used as a principle to choose the right explainer for each model on each dataset. With the best explanation, we use it as the interpretation of the model to conduct interpretability comparisons mentioned in Section 4.2.2.

**Legend:** ■ Negative □ Neutral ■ Positive

| True Label | Predicted Label | Attribution Label | Attribution Score | Word Importance |
|---|---|---|---|---|
| 4 | 4 (1.00) | 4 | 3.21 | node 0 is connected to nodes 15 , 27 , 32 and 37 . node 1 is connected to nodes 2 , 5 , 6 , 8 , 13 , 24 and 36 . node 2 is connected to nodes 1 , 5 , 22 , 24 and 38 . node 3 is connected to nodes 8 , 13 , 19 , 23 and 29 . node 4 is connected to node 27 . node 5 is connected to nodes 1 , 2 , 8 , 12 , 16 , 17 , 27 , 30 , 31 and 37 . node 6 is connected to nodes 1 and 18 . node 7 is connected to nodes 10 , 14 , 22 and 26 . node 8 is connected to nodes 1 , 3 , 5 , 11 , 14 , 23 and 30 . node 9 is connected to nodes 21 and 34 . node 10 is connected to nodes 7 , 12 , 15 , 17 , 33 and 35 . node 11 is connected to nodes 8 , 24 and 26 . node 12 is connected to nodes 5 , 10 , 22 and 30 . node 13 is connected to nodes 1 , 3 , 22 and 23 . node 14 is connected to nodes 7 , 8 , 18 , 19 and 27 . node 15 is connected to nodes 0 , 10 , 22 , 26 , 34 and 37 . node 16 is connected to nodes 5 , 19 , 32 and 36 . node 17 is connected to nodes 5 , 10 , 22 , 27 and 34 . node 18 is connected to nodes 6 , 14 and 32 . node 19 is connected to nodes 3 , 14 , 16 , 22 , 24 and 30 . node 20 is connected to nodes 22 , 27 , 28 , 31 and 32 . node 21 is connected to nodes 9 and 25 . node 22 is connected to nodes 2 , 7 , 12 , 13 , 15 , 17 , 19 , 20 , 23 , 24 , 32 and 34 . node 23 is connected to nodes 3 , 8 , 13 , 22 and 34 . node 24 is connected to nodes 1 , 2 , 11 , 19 and 22 . node 25 is connected to node 21 node 26 is connected to nodes 7 , 11 , 15 and 37 . node 27 is connected to nodes 0 , 4 , 5 , 14 , 17 and 20 . node 28 is connected to node 20 . node 29 is connected to node 3 . node 30 is connected to nodes 5 , 8 , 12 and 19 . node 31 is connected to nodes 5 and 20 . node 32 is connected to nodes 0 , 16 , 18 , 20 and 22 . node 33 is connected to nodes 10 and 38 . node 34 is connected to nodes 9 , 15 , 17 , 22 , 23 and 38 . node 35 is connected to node 10 . node 36 is connected to nodes 1 and 16 . node 37 is connected to nodes 0 , 5 , 15 , 26 and 38 . node 38 is connected to nodes 2 , 33 , 34 and 37 . what is the degree of node 0 ? |

Figure 6: **Interpretation visualization of BERT on the node degree dataset.**

### E.3 INTERPRETATION VISUALIZATION

We present the interpretation results in Figures 5 and 6. As depicted, important tokens are highlighted with a green background. The methods under comparison are required to count the nodes connected to node 0 for making predictions. While our method accurately processes this task, BERT fails to correctly identify the relevant node for degree counting. This discrepancy arises because node 0, consistently presented at the beginning during training, is permuted during testing, causing BERT to misidentify its position. In contrast, our method employs a permutation-invariant approach to graph processing, thereby preserving its high performance during testing.

### E.4 LOCAL BLOCK ANALYSIS

The assessment of the node structure annotation embeddings in HLM-G reveals intriguing insights into the model's encoding capabilities. These embeddings, derived from 1-hop neighborhood information, prompt an inquiry into the model's approach to capturing such local graph structures. Specifically, we investigate the positional and structural awareness exhibited by these embeddings, akin to the strategies employed in GNNs and GTs, where Positional Encoding (PE) (Dwivedi et al., 2022) is a common technique for enhancing model performance. PE assigns similar positional values to nodes in close proximity, reflecting their relative positions within the graph.

To evaluate the positional and structural encoding prowess of HLM-G, we create over 10000 pairs of nodes and analyze the node structure annotation embeddings generated by the lower layers of the model. By comparing these embeddings using cosine similarity, we categorize the pairs into three groups based on their hop distance: 1-hop neighbors, 2-hop neighbors, and neighbors at 3 or more hops.

Table 14: **Cosine similarity of 1-hop and 2-hop neighbors with different numbers of common neighbors.** We see that cosine similarity between 1-hop and 2-hop neighbours is quite high and keeps on increasing with increasing number of common neighbors.

| Common Neighbors | 1-hop Neighbors | 2-hop Neighbors |
|---|---|---|
| 1 | 0.956 | 0.931 |
| 3 | 0.957 | 0.939 |
| 5 | 0.966 | 0.954 |
| 7 | 0.972 | 0.951 |
| 9 | 0.975 | 0.968 |

Table 15: **Similarity for 3-hop neighbors (no common neighbors) based on the difference of structure.** The table suggests that lower block assigns similar embedding to nodes that share a common 1-hop structure.

| Difference of Node Degree | Cosine Similarity |
|---|---|
| 0 | 0.955 |
| 1 | 0.839 |
| 2 | 0.557 |
| 3 | 0.024 |
| 4 | -0.113 |
| 5 | -0.129 |
| 6 | -0.135 |

Table 14 and Table 15 reveals a consistent trend in similarity: embeddings of 1-hop neighbors exhibit higher similarity compared to those of 2-hop neighbors, and likewise for 3+ hop neighbors. Furthermore, we observe that the number of common neighbors between two nodes significantly influences the similarity of their embeddings. A higher number of common neighbors indicates greater positional similarity between the nodes.

In the case of 3+ hop neighbors where no common neighbors exist, we evaluate the role of structural similarity. Here, nodes are considered similar in structure if they share a similar 1-hop neighborhood,

specifically in terms of the number of neighbors. The analysis demonstrates that the greater the difference in 1-hop structure between nodes, the lower the similarity in their embeddings. This suggests that HLM-G effectively encodes 1-hop neighborhood information, assigning higher similarity to nodes that are either positionally or structurally similar.

# F  ABLATION STUDIES

## F.1  POOLING MECHANISMS

In the process of constructing node embeddings from the outputs $(H_{U^{AE}}, H_{U^X})$ of the lower layer, we examine two distinct pooling mechanisms: mean pooling and concatenate pooling.

Mean pooling employs a parameter $\alpha$, which signifies the relative importance attributed to structural information. Specifically,

$$z_v = \text{Pool}(H_{U^{AE}}, H_{U^X}) = \alpha * H_{U^{AE}} + (1 - \alpha) * H_{U^X}$$

Essentially, each neuron within $z_v$ encapsulates both structural and feature information. An $\alpha > 0.5$ indicates a predominance of structural information in the final prediction process, whereas $\alpha < 0.5$ suggests that nodal features hold greater significance.

Concatenate pooling, in contrast, yields node embeddings of doubled dimensionality by concatenating structural and feature embeddings,

$$z_v = \text{Pool}(H_{U^{AE}}, H_{U^X}) = \text{concat}(H_{U^{AE}}, H_{U^X})$$

This approach integrates structural and feature vectors, thereby expanding the representational capacity of the resultant node embeddings. The impact of various pooling ratios ($\alpha$) is systematically evaluated across one node-level, link-level and graph-level real-world datasets.

Table 16: **Performance comparison between mean pooling and concatenate pooling** across node- link- and graph- level datasets. $\uparrow \alpha$ implies more structural information is used for making final predictions. Metric is Accuracy for cora and ROC-AUC for molhiv and PubMed. $\alpha = 0$ implies only node features are used for making final prediction whereas $\alpha = 1$ means that predictions rely entirely on the graph's structure.

| Pooling | $\alpha$ | Cora Node | molhiv Graph | PubMed Link |
|---|---|---|---|---|
| | 0.0 | 86.32 | 73.8 | 95.7 |
| | 0.1 | 87.06 | 74.2 | 94.8 |
| | 0.2 | **88.45** | 75.5 | 96.2 |
| **Mean** | 0.3 | 86.9 | **76.39** | 97.2 |
| | 0.4 | 86.3 | 74.2 | 97.4 |
| | 0.5 | 85.9 | 74.5 | **98.24** |
| | 0.6 | 85.58 | 75.6 | 98.2 |
| | 1.0 | 66.35 | 75.1 | 91.1 |
| **Concatenate** | - | 85.35 | 75.1 | 96.6 |

Table 16 shows that mean pooling generally outperforms concatenate pooling, with $\alpha$ values between 0.1 and 0.5 delivering consistently strong results across all datasets. It's crucial to recognize that $\alpha$ measures the structural relevance in the final model. Our findings suggest that features specific to individual nodes are more significant than broader structural characteristics, especially in citation networks such as Cora, where $\alpha = 0.1$ is optimal. Conversely, for the PubMed link dataset, an $\alpha$ value of 0.5 yields the best performance, reflecting the importance of structural connections in conveying critical information about the relationships between nodes.

## F.2 INPUT PROMPT DESIGN

Various prompt designs can be employed to describe graph structures for language models. While the main paper predominantly used a natural language description focusing on 1-hop neighbors (our Current Graph Description Language, CGDL), it's important to assess whether different prompt styles impact the model's performance. In this ablation study, we explore two additional prompt styles: the Adjacency List Format (Adj-List) and Edge List Format (Edges).

Moreover, we investigate the model's out-of-domain (OOD) capabilities under two scenarios:

- **Cross-Prompt Evaluation:** In this setting, models trained on one prompt design (e.g., CGDL) are evaluated on different prompt designs (e.g., Adj-List or Edges) to test adaptability.
- **Node Token Variability:** We introduce OOD test sets where node identifiers are replaced with random strings of up to four characters, simulating a situation where node tokens during inference differ from those encountered during training.

**Experimental Setup:** We conducted our experiments on the Cycle graph reasoning dataset, where we trained separate models using each of the three prompt designs—CGDL, Adj-List, and Edges. Each model was trained independently using the respective prompt format to ensure it could learn the graph structures as described by that particular design. Following training, these models were evaluated on all three prompt formats, as well as their OOD versions with altered node tokens, resulting in a comprehensive assessment of both in-domain and out-of-domain performance. This setup allowed us to rigorously test the adaptability and robustness of our model under varying prompt styles and node token representations.

Table 17: **In-domain and Out-of-domain performance analysis across different prompt styles and node token variations on cycle dataset.** Performance is measured as accuracy (%).

| Training \ Testing | CGDL | Adj-List | Edges | CGDL-OOD | Adj-List-OOD | Edges-OOD |
|---|---|---|---|---|---|---|
| CGDL | **99.5%** | 52.5% | 54.1% | **96.0%** | 51.9% | 53.6% |
| Adj-List | 93.2% | **98.5%** | 74.2% | 73.2% | **94.2%** | 66.5% |
| Edges | 94.5% | 86.0% | **99.0%** | 89.5% | 78.1% | **98.7%** |

**Key Observations:**

1. **Strong In-Domain Performance:** The diagonal entries in Table 17 (99.5%, 98.5%, and 99.0%) indicate that each model performs exceptionally well when evaluated using the same prompt style as the one it was trained on, demonstrating strong in-domain performance. This suggests that our model is capable of effectively learning graph structures regardless of the chosen prompt style.

2. **Resilience to Node Token Variability:** When examining the OOD results where node tokens were changed (CGDL-OOD, Adj-List-OOD, Edges-OOD), each model retained considerable accuracy compared to its in-domain results. For example, the model trained on the Edges format maintained a high performance of 98.7% in the Edges-OOD setting. This suggests that the model is robust against variations in node tokens and can maintain its graph structure understanding even when faced with different node representations.

3. **Superior Generalization with Edge Descriptions:** The model trained with the Edges format demonstrated remarkable generalization ability across both cross-prompt settings and OOD scenarios. It achieved high accuracy when tested on different prompt designs (e.g., 94.5% on CGDL and 86.0% on Adj-List), and similarly performed well even when node tokens were altered. This indicates that training on the Edges format enables the model to adapt more effectively to variations in graph description languages and node representations, making it a versatile choice for different graph tasks.

Overall, this ablation study on input prompt design reveals that our model can handle different input prompt designs and adapt to node token variations, showcasing its strong generalizability and robustness in capturing graph structures across diverse graph description languages.

### F.3 LOCAL BLOCK DESIGN

In this section, we examine different architectural approaches for the local block of our model, focusing on how structure and node features are processed. Traditionally, these features are handled hierarchically, meaning they are processed independently from each other. The input to the hierarchical design lower block is structured as follows:

$$U_G = (\text{concat}(U_1^X, U_1^{AE}), \text{concat}(U_2^X, U_2^{AE}), \ldots, \text{concat}(U_n^X, U_n^{AE}), U_Q)$$

This approach employs a single lower block, $M_L$, which processes the concatenated features hierarchically.

Following the hierarchical model, we introduce a double hierarchical design, which further divides the handling of node and structural features. In this enhanced setup, we implement two distinct lower blocks: $M_{L_1}$ for node features and $M_{L_2}$ for structural features. The input for this double hierarchical design is given by:

$$U_G = \text{concat}(U^{AE}v_1, U^X v_1, \cdots, U^{AE}v_n, U^X v_n, Q_G)$$

This arrangement allows each lower block to specialize, thereby enhancing their processing capabilities on their respective feature types.

Table 18: **Comparison of Model Performance by Design Configuration.** Accuracy is used as the metric. This table presents performance metrics across different datasets, distinguishing between Hierarchical and Double Hierarchical design models.

| Dataset | Type | Double Hierarchical Design | | Hierarchical Design |
|---|---|---|---|---|
| | | 1 Lower Block | 2 Lower Blocks | 1 Lower Block |
| Pubmed | Node | **94.25** | 93.9 | 92.9 |
| Cora | Node | 87.8 | **88.3** | 86.1 |
| WN18RR | Link | **97.6** | 97.5 | 97.3 |

From Table 18, we note a slight performance advantage with the double hierarchical design. This design enables the model to more effectively differentiate between node features and structural elements, as these are processed independently in the input, leading to improved performance. The double hierarchical design exhibits comparable results whether using one or two lower blocks. Given the similar performance outcomes, we opt for a single lower block due to its lower parameter count—using two blocks would nearly double the parameters from 86M to 150M. Therefore, in scenarios where parameter efficiency is critical, the double hierarchical design with a single lower block is preferable.

## G LIMITATIONS, CHALLENGES, AND PERSPECTIVES

### G.1 LIMITATIONS

The most significant limitation of our current methodology lies in its lack of zero- and few-shot learning capabilities. Recent advancements in Large Language Models (LLMs) have shown exceptional proficiency in zero- and few-shot scenarios, suggesting an urgent need for research aimed at integrating these capabilities into our approach. An initial attempt to address this, described in Appendix D.2, involves using our model as an encoder coupled with a powerful LLM decoder through prefix tuning. While this approach enhances fine-tuning efficacy, it falls short in generalizing few-shot abilities.

Powerful decoder based LLMs can be used in the future leveraging a similar local to global architecture (using similar attention masks). Earlier layers can be set to focus on tokens of the same node (mimicking intra-node attention). Due to the Causal attention used in decoder LLMs, the last token in every node's description can be either directly be used as the node token in upper block or after pooling with other tokens of same node, mimicking inter-node attention. However, more research is needed and we leave this to future work.

## G.2 CHALLENGES

A notable challenge in enhancing our model involves rethinking the attention mechanisms employed in LLMs. Our model benefits from a unique local and global attention scheme, which could inform modifications to the attention masks in LLMs. For example, adapting Transformer block architectures within LLMs to split the layers into two distinct blocks—one focusing exclusively on prior tokens of the current node (lower block) and the other emphasizing a single embedding for every node (upper block)—could be a strategy. However, this structural modification is complex to code and train on LLMs, and it demands substantial computational resources and algorithmic innovation.

## G.3 PERSPECTIVES

Hybrid models that combine the structure analysis of Graph Neural Networks (GNNs) with the language skills of Large Language Models (LLMs) show great promise for creating stronger systems. These models use the broad abilities of LLMs to work well across different areas, helping to overcome the specific limitations of traditional GNN architectures. Such models are suited for a wide range of graph-related tasks in real-world settings, compensating for the limitations of GNNs, which usually have only a few million parameters and don't always perform consistently across different fields. This issue highlights the need for better encoding mechanisms that can represent graph data effectively, whether it's for knowledge graphs, molecular structures, or social networks.

Despite increasing interest and some early successes, there are still major challenges, especially in making these models work well across very different areas. Most current research focuses on node classification tasks, which don't fully show what these hybrid models can do in broader applications. Additionally, tests of these models on various graph reasoning tasks are rare, and the results haven't yet shown major breakthroughs. This points to a clear need for more focused research to truly understand these models' abilities to interpret complex structures, identifying it as a key area for future developments.

In conclusion, while our model introduces innovative solutions to graph data analysis, the path forward involves addressing its scalability to zero-shot learning, enhancing its integration with LLM architectures, and expanding its adaptability to diverse and complex graph structures. These developments will not only advance the theoretical foundations of graph neural networks but also expand their applicability and effectiveness in practical scenarios.

