# OpenReview forum: "A Hierarchical Language Model Design For Interpretable Graph Reasoning"
_ICLR.cc/2025/Conference — Submitted to ICLR 2025_

### Official Review · Reviewer_ju28 · 2024-10-29

**Soundness:** 2
**Presentation:** 3
**Contribution:** 2
**Rating:** 3
**Confidence:** 4

**Summary:**

The paper proposes HLM-G to enhance the graph structure understanding of a language model. Specifically, HLM-G introduces two blocks: the local block and the global block to capture information from node-specific structures and features, as well as global graph level, respectively. A pooling layer is also adopted between the two blocks to integrate structural and feature-based information extracted from the graph. Extensive experimental results show the effectiveness of the proposed model.

**Strengths:**

1.	The paper is overall well-structured, and the logic of the paper is easy to follow.
2.	The templates/examples the authors gave such as the node feature/structure annotations are reader-friendly, and are effective to help readers have a direct understanding how graphs are described in natural language.
3.	The hierarchical design of the model looks very straightforward to the readers and the demonstration of the model in figure 1 is very clear.
4.	As the authors turn the graphs into natural language description, it somehow enhances the interpretability of the model in human language aspect.

**Weaknesses:**

1.	The design of natural language descriptions of graphs is very rigid. Personally, I don’t buy in this kind of language descriptions of the graphs according to my own experience.
2.	As the authors mention that “the node feature annotation for a node is a natural language sequence that describes the attributes over a predefined vocabulary”, which indicates that the model can only be applied to text-attributed graphs, but no other kinds of the graph, which narrows the use of the model.
3.	Lack key ablation studies on the blocks. The ablation experiments in the appendix actually don’t answer the questions of how each of the component in your framework contribute to the final performance. I’d like to see the experimental results with local block or global block only to see if these designs are really helpful.
4.	To be honest, the synthetic graph datasets with up to 40 nodes and 700+ edges in graph reasoning are too small, and the experimental results are not persuasive. Although it is mentioned in the appendix that currently available datasets (e.g., NLGraph and GraphQA Benchmark) suffer from category imbalance and low number of nodes (around 10-20), these random synthetic graphs are still unconvincing. It is not possible to adequately demonstrate the performance of the model on real-world graph data.
5.	Incomplete experimental comparison. In the experiments on real world datasets, the model is mainly compared with the zero/few samples setting of traditional GNN and LLM, and the coverage of the experiments is not comprehensive enough. For example, lacking LLaGA as mentioned in the existing work, OFA is only demonstrated in the link prediction task, lacking a broader comparison. This makes it difficult to fully assess the competitiveness of the model against state-of-the-art techniques.
6.	I am doubt about the interpretability, and the approach of reflecting the interpretability of the model through the distribution of attentional weights is somewhat controversial. Whether the attentional weights can truly reflect the model's understanding of the graph structure is still an open question, as high weights do not always imply the importance of the features to the final decision.
7.	When dealing with graphs with larger size, nodes are divided into different batches, each of which contains only 40 nodes in the paper, which make information difficult to flow over different batches, especially for the global block. (Actually there is even no interaction between different batches!)

**Questions:**

1.	In the abstract, you mention that “we introduce a Hierarchical Language Model Design”, but I don’t think it is a language model design, instead it is more likely a framework where you can use different LLMs as the backbone model, isn’t it?
2.	Is the word “hierarchical” appropriate? According to my understanding, the main role of Local block is to process the textual features and neighbor connectivity of a node through Transformer and encode it into a node vector through pooling operation. The Global block, on the other hand, after integrating these node vectors, is further processed by the Transformer to capture the entire subgraph representation. Such a processing flow I think is not consistent with what we often say about considering the local structure of nodes and the global structure of remote nodes.
3.	How can the model be applied to those non-text-attributed graphs?
4.	How can the model have a global understanding of the graph since you separate nodes into many batches? What is the specific design when dealing with large graphs?
5.	How to deal with the length difference of the feature and structure annotations of different nodes?

---

### Official Review · Reviewer_aRpH · 2024-10-31

**Soundness:** 2
**Presentation:** 3
**Contribution:** 2
**Rating:** 5
**Confidence:** 4

**Summary:**

This paper proposes a Hierarchical Language Model for Graphs (HLM-G) to enhance the graph structure comprehension capabilities of LLMs. Specifically, HLM-G employs a two-block architecture, comprising a local block and a global block to capture graph features and structures, which not only enhances the model’s understanding of graph tasks but significantly reduces computational costs. Experiments show HLM-G achieves the better performance compared to existing baselines.

**Strengths:**

Advantages
1. The proposed method is technically sound, and it has some novelty.
2. The paper is easy to follow and well organized.
3. The experiments are comprehensive, including overall performance, robustness assessment, interpretation comparison and efficiency analysis.

**Weaknesses:**

Disadvantages

1. The motivation behind the paper is not clearly explained.
2. Section 3.1 presents the reformulation of graph-level task, describing the graph structure associated with each node. However, this approach may contain redundant information.
3. The ablation studies does not seem to explore the effect of local and global blocks on model performance, respectively.
4. Section 4.2 employs ground truth ranking to evaluate the model's interpretability; however, this approach seems imprecise. The ranking more accurately reflects the model's effectiveness rather than its interpretability.
5. The paper is not rigorously expressed. For example, the variable subscripts are inconsistent, and the variables “N” and “n” are used confusingly in Section 3.

**Questions:**

Existing methods faces two challenges: handling graph structures and addressing scalability issues. It is unclear how HLM-G addresses these challenges using local and global blocks.

---

### Official Review · Reviewer_Ac8p · 2024-11-03

**Soundness:** 4
**Presentation:** 4
**Contribution:** 3
**Rating:** 6
**Confidence:** 4

**Summary:**

The paper introduces a text-based graph encoder designed to handle both node and edge features. The proposed architecture consists of two primary components: initially, each node is processed individually, resulting in two embedding vectors per node. These vectors are then combined through a weighted average, after which an attention-based mechanism is used to propagate information across nodes. Finally, an MLP predicts the label associated with a given query (e.g., node count or cycle existence).

**Strengths:**

The manuscript is highly detailed and well-structured. The appendix, in particular, offers extensive supplementary information, which will likely be valuable to researchers in this domain.

The authors validate their approach on a wide array of both real and synthetic datasets and provide a robust baseline comparison, including traditional graph neural networks (GNNs) and models based on LLMs. Notably, the model presents a degree of interpretability, a compelling feature given the typically opaque nature of black-box models.

**Weaknesses:**

1. **Relation of HLM-G to LLMs.** The paper associates HLM-G with large language models (LLMs). However, based on the text, it appears that HLM-G is trained separately and from scratch on each task without any apparent connection to pre-trained LLMs (unlike, for instance, GraphToken). This association could be misleading, as there is also no indication that HLM-G inherits the capabilities typically attributed to LLMs. Given HLM-G’s merits as a graph encoding method, it stands as a valuable contribution on its own independent of the LLM association, and this should be reflected in the manuscript too.

2. **Performance on Baseline Comparisons.** While HLM-G demonstrates impressive performance on synthetic datasets (Table 1), this superiority does not extend as clearly to real-world datasets (Tables 3-5), where its performance closely aligns with simpler models like GCN. Insights from the authors on this discrepancy could shed light on the strengths and limitations of HLM-G in practical settings.

**Questions:**

1. **Parameter Count.** While HLM-G performs strongly across Tables 1-5, the number of trainable and frozen parameters for each baseline model is not specified. Including this information would facilitate a more balanced and fair comparison.

2. **GPU Usage and Computational Requirements** The authors note the use of A6000 GPUs but do not specify the number required to replicate the experiments in Tables 12 and 13. Additionally, these tables feature only the LLM-based models, potentially creating an incomplete picture of computational needs. Adding a comparison with GNN-based models would offer readers a clearer perspective on resource demands.

3. **Formulating Queries for GNN Encoders.** Given that GNNs are not text-based encoders, clarification on how queries (e.g., for node degree) are formulated for these models would help contextualize the comparative results and better explain the huge performance difference.

---

### Official Review · Reviewer_SCB1 · 2024-11-04

**Soundness:** 3
**Presentation:** 2
**Contribution:** 2
**Rating:** 5
**Confidence:** 4

**Summary:**

This work presents the Hierarchical Language Model (HLM-G) Design, which employs a two-block architecture to enhance graph structure understanding. The model achieves state-of-the-art performance, outperforming both GNN and LLM baselines, and shows robustness to variations in graph prompts. Its interpretability is demonstrated through attention weights and established explainers. Evaluations across diverse real-world datasets highlight the model’s superior generalization capabilities, marking a significant advancement in applying LLMs to graph-centric tasks.

**Strengths:**

The structure is simple and easy to understand. It considers improving the reasoning capabilities of LLMs from the perspective of computational efficiency, which is a promising aspect.

**Weaknesses:**

1. The lines 040-058 of the article introduce two important challenges: the first is the challenge of obtaining structural information, and the second is the challenge of scalability. However, regarding challenge two, the strategy proposed in lines 060-072 does not provide me with a clear understanding of how this challenge is addressed.
2. From the design of the article, it is not clear how the ability of LLMs to understand graph structures is enhanced, as referenced in challenge 1. Moreover, according to Figure 1, I cannot even identify the role played by LLMs within it.
3. In the comparative analysis of interpretability performance, the article uses tasks such as shortest distance, reachability, edge existence, and node degree. However, the main downstream tasks used for comparison are graph classification and node classification tasks.
4. On page 20, in the Task 1 section for Shortest Distance, the range is only 0-5, and experiments with path graphs are missing. Clearly, the paths in path graphs are longer.
5. On page 19, the first line states, "In comparison, our method is inherently task-agnostic and demonstrates high interpretability." Honestly, I do not find this model to be particularly model-agnostic, and there is a lack of stronger experimental evidence to support its interpretability claims.
6. In the last sentence of the final paragraph on page 30, it states, "This indicates that HLM-G can effectively encode 1-hop neighborhood information, assigning higher similarity to nodes that are similar in position or structure." However, there is no explanation of the impact of 2-hop encoding on the nodes.
7.  On page 30, the first line states, "This discrepancy arises because node 0, consistently presented at the beginning during training, is permuted during testing, causing BERT to misidentify its position." This suggests that node 0's position in the training set is always first, leading to poor performance when its order is changed in the test set. However, Figure 6 does not reflect this issue in BERT's test results.

**Questions:**

1. Line 179 states, "Since language models cannot inherently understand graphs in their natural structure." What is the basis for this statement?
2. What is the design intuition behind separately handling structural information and feature information of nodes? It is well known that GNNs process both aspects of information simultaneously.
3. Further supplement the experiments on interpretability performance comparison to include both graph classification and node classification tasks.

---

### Meta-Review · Area_Chair_FazU · 2024-12-20

**Metareview:**

This paper presents a text-based graph encoder capable of processing both node and edge features. The architecture is built around two main components: first, each node is independently encoded to produce two embedding vectors per node. These embeddings are subsequently merged using a weighted average. Next, an attention-based mechanism facilitates information propagation between nodes. Finally, a multi-layer perceptron (MLP) predicts the label corresponding to a specific query, such as node count or the presence of a cycle.

Although the reviewers found the paper interesting and easy to follow, there are several major concerns regarding the clarity of the motivation, completeness of the experimental comparison, and sufficiency of the analysis. The authors didn't provide a response, and thus I am recommending rejection of this work.

**Additional Comments On Reviewer Discussion:**

There is no rebuttal from the authors.

---

### Decision · Program_Chairs · 2025-01-22

Reject